# Discovery of levodopa-induced dyskinesia-associated genes using genomic studies in patients and *Drosophila* behavioral analyses

Woongchang Yoon [1,4,6], Soohong Min[1,5,6], Ho-Sung Ryu[2], Sun Ju Chung[3] & Jongkyeong Chung [1✉]

Although levodopa is the most effective medication for Parkinson's disease, long-term levodopa treatment is largely compromised due to late motor complications, including levodopa-induced dyskinesia (LID). However, the genetic basis of LID pathogenesis has not been fully understood. Here, we discover genes pathogenic for LID using *Drosophila* genetics and behavioral analyses combined with genome-wide association studies on 578 patients clinically diagnosed with LID. Similar to the therapeutic effect of levodopa in patients, acute levodopa treatments restore the motor defect of Parkinson's disease model flies, while prolonged treatments cause LID-related symptoms, such as increased yawing, freezing and abrupt acceleration of locomotion. These symptoms require *dopamine 1-like receptor 1* and are induced by neuronal overexpression of the receptor. Among genes selected from our analyses in the patient genome, neuronal knockdown of *adenylyl cyclase 2* suppresses the levodopa-induced phenotypes and the receptor overexpression-induced symptoms in *Drosophila*. Together, our study provides genetic insights for LID pathogenesis through the D1-like receptor-adenylyl cyclase 2 signaling axis.

[1] School of Biological Sciences and Institute of Molecular Biology and Genetics, Seoul National University, 1 Gwanak-Ro, Gwanak-Gu, Seoul 08826, Republic of Korea. [2] Department of Neurology, Kyungpook National University Hospital, 130 Dongdeok-Ro, Jung-Gu, Daegu 41944, Republic of Korea. [3] Department of Neurology, Asan Medical Center, University of Ulsan College of Medicine, 88 Olympic-Ro 43 Gil, Songpa-Gu, Seoul 05505, Republic of Korea. [4] Present address: Bio-MAX/N-Bio Institute, Seoul National University, 1 Gwanak-Ro, Gwanak-Gu, Seoul 08826, Republic of Korea. [5] Present address: Department of Cell Biology, Harvard Medical School, 240 Longwood Avenue, Boston, MA 02115, USA. [6] These authors contributed equally: Woongchang Yoon, Soohong Min. ✉email: jkc@snu.ac.kr

Parkinson's disease (PD) is a common neurodegenerative movement disorder characterized by the progressive loss of dopaminergic neurons in the substantia nigra of the midbrain and ~~the~~ additional neurological traits in other brainstem neurons, including serotonergic, cholinergic, and noradrenergic neurons[1,2]. The symptomatology of PD is heterogeneously composed of classical motor features of resting tremor, bradykinesia, rigidity and gait disturbance, along with non-motor features[3]. Despite these complicated symptoms, oral intake of levodopa (L-DOPA) has been the most effective medication for decades to treat the disease[4]. However, the therapeutic effect by L-DOPA is largely compromised due to the occurrence of late motor complications, termed L-DOPA-induced dyskinesia (LID) caused by the long-term treatment of L-DOPA[5]. LID represents a spectrum of abnormal involuntary movements (AIMs) including chorea (quick involuntary movements of the limbs), dystonia (involuntary twisting movements or abnormal fixed posture), and athetosis (involuntary writhing movements)[6–8]. The vast majority of PD patients eventually develops one of these LID symptoms several years after L-DOPA treatment and is challenged in their quality of life. Almost ninety percent of PD patients suffer from LID nine years after L-DOPA treatment[9]. Initial LID symptoms usually occur at the peak level of L-DOPA in the plasma, but as L-DOPA therapy persists, LID may appear even when its level is arising or declining in the form of diphasic dyskinesia[10]. These suggest that chronic L-DOPA application causes adaptation processes that involve pathogenic changes in the dopamine signaling pathway[11]. In order to delay the onset of LID, minimal dosing of L-DOPA in combination with carbidopa or late initiation of L-DOPA treatment is taken as an alternative approach[12]. However, once LID has developed, it is irreversible and termination of L-DOPA treatments worsens the motor symptoms, making LID difficult to control[13]. Therefore, studying the pathogenic alterations in the dopamine signaling system upon prolonged L-DOPA treatment is imminent to provide therapeutic implications for LID[14].

There are two different views on the pathogenic mechanism underlying LID: presynaptic vs postsynaptic mechanism in the dopamine signaling system[15]. Chronic dosing of L-DOPA supplies non-physiological levels of dopamine in the presynaptic terminals, and in turn can override the endogenous capacity of dopamine clearance, leading to fluctuations in dopamine levels that can evoke involuntary motor symptoms[16]. Abnormalities in dopamine transporters and auto-receptors that regulate conversion and reuptake of dopamine have been shown to correlate with LID in PD patients[17]. In addition to these presynaptic changes, pulsatile supply of excessive dopamine in the synaptic cleft can cause abnormal activity/expression of dopamine receptors in the postsynaptic neurons, resulting in irreversible alterations in downstream signaling molecules that control transcription and translation[14]. These molecular alterations in the postsynaptic neurons could eventually impact functional and structural facets of the neurons that govern motor output[8]. Indeed, a wide range of dysregulations in cellular signaling is associated with LID pathogenesis[18]. Lines of evidence have shown that D1-like family receptors and their downstream signaling are important for LID pathogenesis[19]. D1-like family receptors include dopamine receptor D1 (DRD1) and dopamine receptor D5 (DRD5) coupled to Gsα that stimulates adenylyl cyclases (ADCYs) for induction of cyclic adenosine monophosphate (cAMP)[20]. Dopamine receptor D2 (DRD2), dopamine receptor D3 (DRD3), and dopamine receptor D4 (DRD4) belong to D2-like family receptors that bind to Giα which inhibits cAMP signaling[21]. In particular, cAMP-associated signaling components, such as ADCYs, protein kinase A (PKA) and dopamine and cAMP-regulated phosphoprotein of 32 kDa (DARPP-32) are implicated in LID[22]. There have been many attempts to examine the role of these molecules in LID pathogenesis by manipulating them genetically and pharmacologically[13,23]. However, natural variant mutations on these molecules linked to PD patients diagnosed with LID have been less understood.

Autosomal dominant mutations on *adenylyl cyclase 5* (*ADCY5*) are implicated in familial dyskinesia and knocking out of *ADCY5* ameliorated the LID symptoms of L-DOPA-treated PD model mice[24]. Considering varying LID responses to L-DOPA treatment, there could be risk factors that genetically make PD patients vulnerable to LID pathogenesis[25]. In an effort to identify such a genetic factor, we established a large collection of single nucleotide polymorphisms (SNPs) from PD patients (>500,000) who developed LID. To screen one that is related to LID symptoms, we sought a model organism that enables a high-throughput screen with preserved motor symptoms upon L-DOPA treatment. In *Drosophila*, artificial supplement of dietary L-DOPA rescues the motor symptoms of flies that lack dopamine synthesis pathway in the nervous system[26,27]. L-DOPA treatment produces therapeutic effects on the motor deficits of a PD model fly line, demonstrating that L-DOPA administration impacts fly motor control via dopamine signaling system[28]. Furthermore, dopamine pathways are preserved anatomically and molecularly in the fly with simplicity and accessibility permitted by fluent genetic tools[29]. Thus, *Drosophila* provides a highly tractable system for studying the genetic mechanism of L-DOPA-induced motor symptoms.

Here, we combined genome-wide association studies (GWAS) on LID-diagnosed PD patients with *Drosophila* genetic tools to investigate LID pathogenesis. First, we established a behavioral paradigm to analyze L-DOPA-induced alterations in fly locomotion. Then, we characterized three components in fly locomotion, speed, yawing and freezing that are statistically correlated with the dose and duration of L-DOPA treatment. In particular, flies with L-DOPA treatment showed abrupt peaks of accelerated movements that deviated from the nearest time points. Using a scoring algorithm, we defined the peaks as AIMs, indicative of LID along with yawing and freezing behaviors. Remarkably, prolonged L-DOPA treatment markedly elevated AIM scores in PD model flies, while acute L-DOPA treatment improved the motor defects of the flies with no sign of AIMs. Neuronal knockdown of *dopamine 1-like receptor 1* (*Dop1R1*, orthologous to *D1-like receptor*) suppressed LID-like symptoms involving pathological elevation of AIM scores, whereas neuronal over-expression of *Dop1R1* was sufficient to trigger the LID-like symptoms even without L-DOPA treatment. Using the fly genetic tools, we performed an RNAi-based genetic screen on the conserved fly genes that were selected from the SNPs of PD patients diagnosed with LID. The screen revealed that knockdown of *Drosophila adenylyl cyclase 2* (*dADCY2*) blocked L-DOPA-induced pathological symptoms. Interestingly, the LID-like symptoms caused by neuronal *Dop1R1* overexpression were also completely suppressed by knockdown of *dADCY2*. Based on these data, we propose that the D1-like family receptor-ADCY2 signaling axis underlies LID pathogenesis.

## Results

**L-DOPA diet influences locomotor parameters in *Drosophila*.** To investigate the effects of L-DOPA intake in fly locomotion, we developed a behavioral paradigm where flies were initially fed on L-DOPA-containing food and transferred to an arena equipped with an automated video tracking system for computational behavioral pattern analyses (Fig. 1a). Wild type (WT) flies on L-DOPA diet displayed abnormal locomotor patterns in response to all concentrations tested (Fig. 1b). For subsequent experiments,

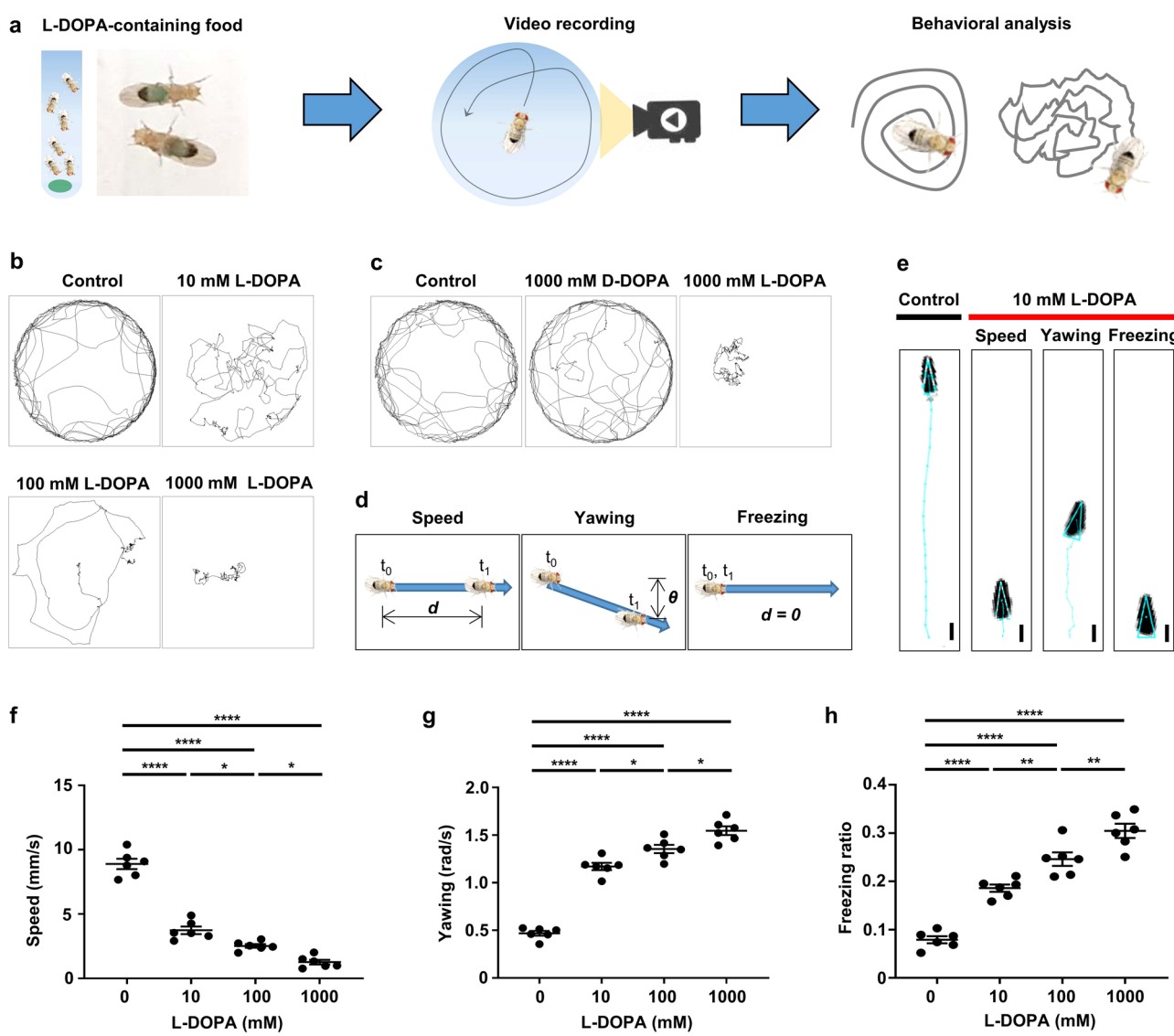

**Fig. 1 L-DOPA treatment alters locomotor parameters of *Drosophila*. a** Experimental scheme to analyze single fly trajectories upon L-DOPA diet using a fly movement tracking system. **b**, **c** Representative trajectories of flies on control and L-DOPA diet with indicated concentrations (**b**), and flies on D-DOPA vs L-DOPA diet (**c**). **d** Schematic illustrations to define speed, yawing and freezing as locomotor parameters. **e** Representative trajectories illustrating speed, yawing, and freezing extracted from 20 frames of movement videos of a control and an L-DOPA-fed fly. One frame corresponds to 0.033 seconds. Bars indicate 1 mm. **f–h** Comparisons of the quantified speed (**f**), fraction of freezing (**g**), and yawing (**h**) of the flies on L-DOPA diet with indicated concentrations. $N = 6$. ****$p < 0.0001$; **$p < 0.01$; *$p < 0.05$ by one-way ANOVA Tukey's multiple comparison test. Data are presented as means ± SEM. $p < 0.05$ was considered statistically significant.

we used 10 mM concentration to avoid potential overdosing effects in flies[4]. We asked behavioral specificity induced upon L-DOPA diet by examining flies on a diet with dextrodopa (D-DOPA), the biologically inactive isomer of L-DOPA. We observed that behavioral abnormalities were caused by L-DOPA, but not by D-DOPA (Fig. 1c). Flies were viable upon prolonged periods of L-DOPA diet comparable to control and D-DOPA diet (Supplementary Fig. 1). To establish reliable quantitative measures that would capture L-DOPA-induced motor complications, we calculated three values: speed, yawing and freezing (Fig. 1d and see Methods). These values have been widely used to establish both *Drosophila* and mammalian models of LID[30–32]. We infer those irregularities in fly speed, orientation, and freezing upon L-DOPA treatment could represent some of the features of LID in mammals such as involuntary movements of the limbs, writhing, and abnormal fixed posture[33]. The values of speed, yawing, and freezing were calculated from distance moved ($d$),

the angular difference between the two orientations ($\theta$), and the fraction of time with zero distance moved ($d = 0$) during the total time period ($t_0$ to $t_1$), respectively (Fig. 1d). L-DOPA intake in flies produced 58.33% decrease in speed, 249.79% elevation of yawing, and 235.32% increase of freezing correlated with L-DOPA concentrations (Fig. 1e–h).

**L-DOPA treatment induces AIMs in *Drosophila*.** The hallmark of LID symptoms in humans is the spectrum of hyperkinetic movements, including chorea, dystonia and myoclonus, which appear abruptly with rapid movements of body parts in an uncontrolled fashion from a stationary posture[6,7]. In the rodent model of LID, these hyperkinetic movements are collectively translated into the abnormal involuntary movement (AIM) as a well-established LID parameter[8,34]. To establish the *Drosophila* version of AIM, we sought to characterize the abruptly

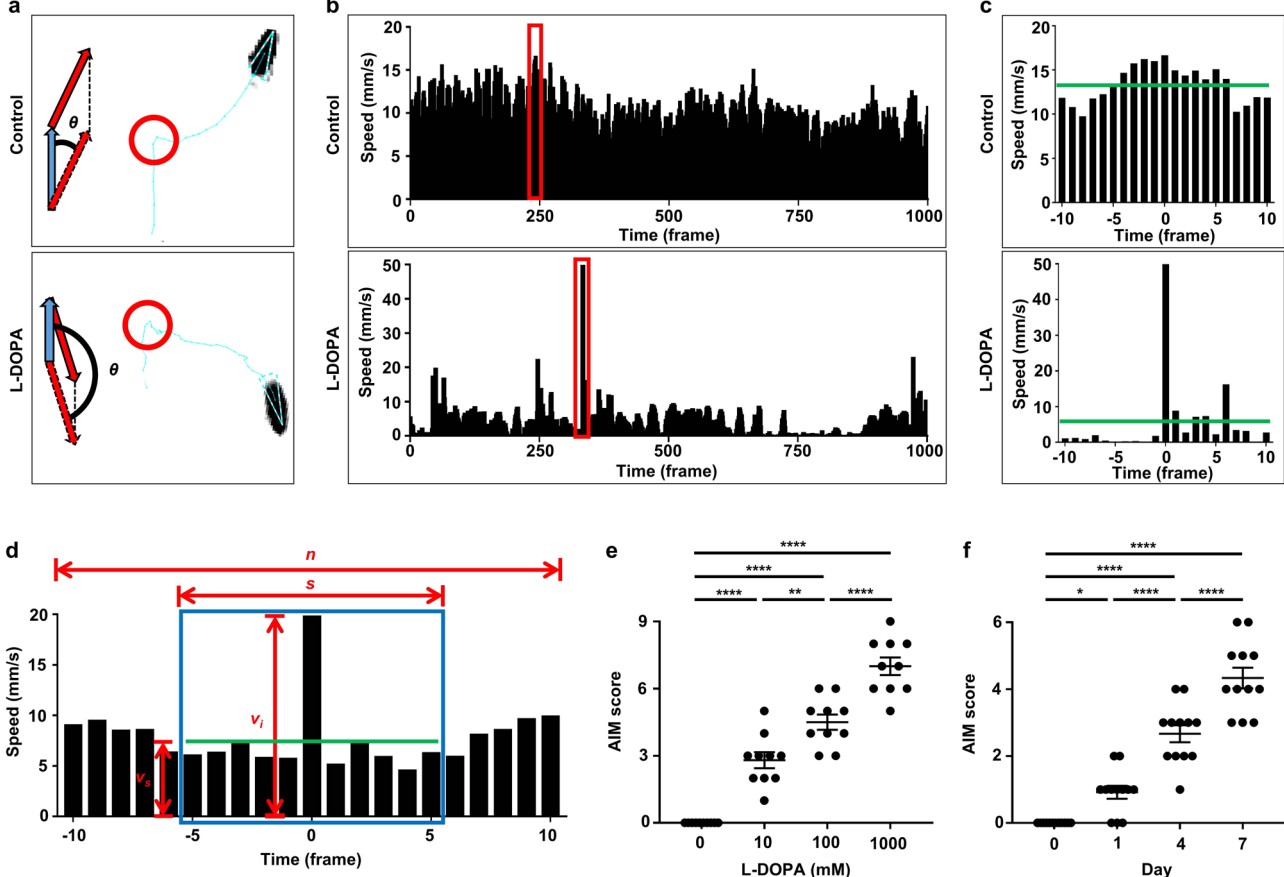

**Fig. 2 L-DOPA treatment induces AIMs in *Drosophila*. a** Representative trajectories of a control and an L-DOPA-fed fly highlighted with their altered directions (red arrows) in comparison to original directions (blue arrow). Red circles depict turning movements. θ indicates the angular difference between the original and altered direction. **b** Representative instantaneous speeds of a control and an L-DOPA-fed fly within 1000 frames. Peaks of instantaneous speeds of the control and L-DOPA-fed fly **c** are highlighted in red boxes of **b**. **c** Zoomed-in instantaneous speeds 10 frames before and after the peaks. Green lines denote the mean speed within 21 frames. **d** Graphical illustration of the variables from instantaneous speeds to calculate the AIM score: s, specific time period (window); vs, the mean speed in the window; vi, instantaneous speed; n, entire recording time period. Green line indicates the mean speed in the window. **e**, **f** Comparisons of the quantified AIM score by indicated concentrations of L-DOPA (**e**) and time periods of L-DOPA diet (**f**). $N = 10–12$. ****$p < 0.0001$; **$p < 0.01$; *$p < 0.05$ by one-way ANOVA Tukey's multiple comparison test. Data are presented as means ± SEM. $p < 0.05$ was considered statistically significant.

accelerated fly movements that are highly irregular and deviated from the regular movement at the near time point upon L-DOPA treatment. Using computational analysis, we systematically calculated instantaneous speeds and deviations in the fly movement, and indeed found that L-DOPA diet caused abrupt accelerations of fly movements upon making a directional change associated with larger angular changes (Fig. 2a). Within a thousand frame, a control fly showed a maximal speed 21.14% deviated from the regular speed, while the L-DOPA-fed fly exhibited a peak of abruptly accelerated speed with 832.07% deviation (Fig. 2b, c). We systematically analyzed this hyperkinetic movement during the entire recording period as AIMs. In a formula, instantaneous speeds ($v_i$) were divided by mean speeds ($v_s$) within certain time periods defined as windows ($s$). The sliding window was used to obtain serial values of $v_i$ over $v_s$ ($v_i/v_s$) within the entire recording time ($n$). Then, the serial $v_i/v_s$ values were logarithmized to examine whether a given $log(v_i/v_s)$ was qualified for a threshold value ($c$) that had been previously determined by WT flies on control diet. The AIM score ($H$) is the sum of 0 or 1 given to each frame (9018 frames total/ 5 min) during the entire recording based on whether the value of $log(v_i/v_s)$ at a frame qualifies the threshold value ($c$). If the value of $log(v_i/v_s)$ is larger than $c$, 1 is given to the frame. If the value of $log(v_i/v_s)$ is smaller than $c$, 0 is

given. For example, if a fly showed the value of $log(v_i/v_s)$ larger than $c$ at seven different frames, the AIM score of the fly is 7. For better visualization of data, the AIM score of control flies is normalized to be 0, where the threshold value ($c$) is 0.4 (Fig. 2d, Supplementary Note 1 and see Methods). Based on this algorithm, we observed that AIM scores were statistically correlated with L-DOPA concentrations and duration of L-DOPA diet (Fig. 2e, f).

**Acute L-DOPA diet rescues defects of PD model flies, but the prolonged diet induces AIMs.** Next, we addressed L-DOPA-induced impacts on PD model flies. Flies lacking *PTEN-induced kinase 1* (*PINK1*) are well-established PD model animals, reca-pitulating the primary symptoms of PD[35]. To monitor PD symptoms affected by L-DOPA, we designed a four-day video-recording experiment intercalated with one-day L-DOPA diet sequences (Fig. 3a). On day 0, *PINK1* null flies (*PINK1^B9^*) without L-DOPA diet showed retarded movements and speed with minimal changes in AIM scores comparable to revertant flies (*PINK1^RV^*) (Fig. 3b–d, Supplementary Fig. 2a, b). On day 1 with L-DOPA diet, the PD-associated motor defects of *PINK1^B9^* flies were fully rescued with no significant changes in AIM scores

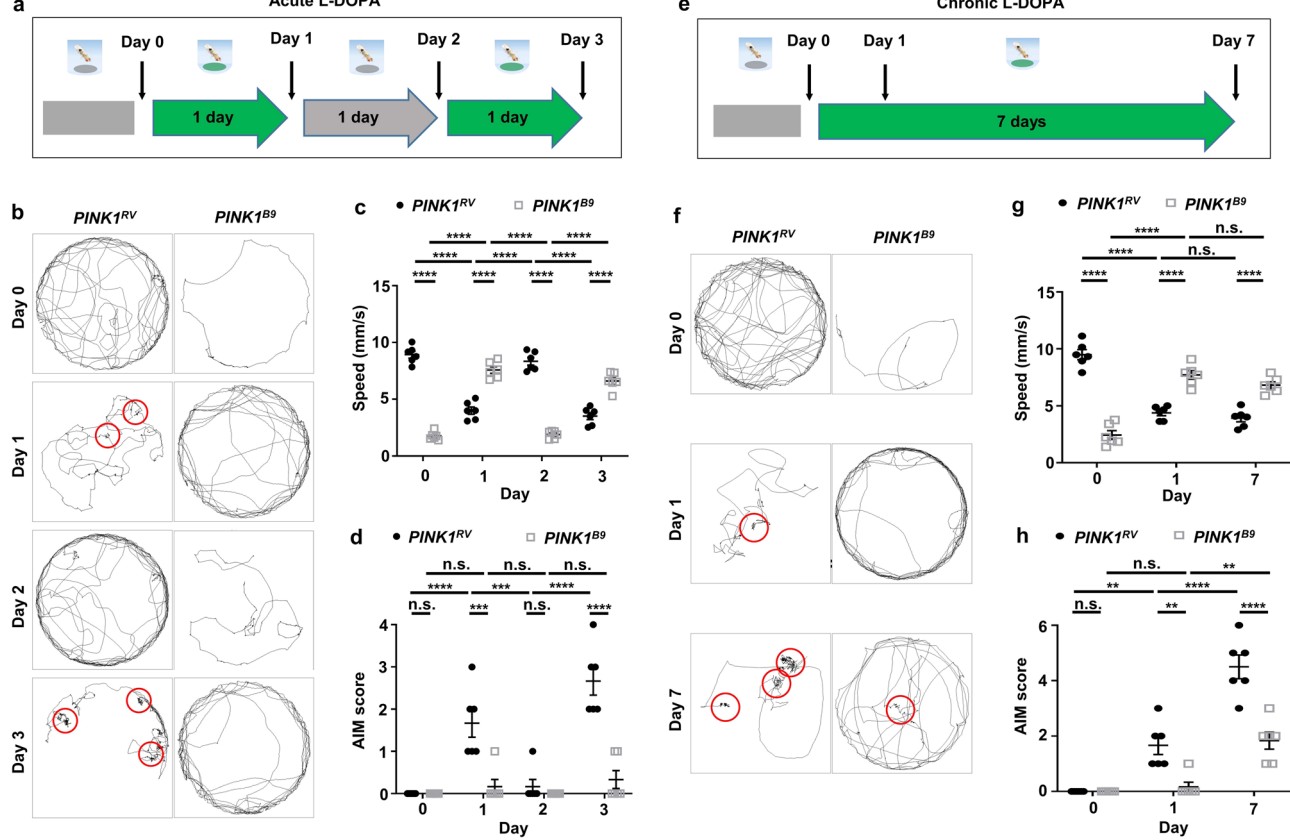

**Fig. 3 Acute L-DOPA treatment rescues PD-associated motor defects, but prolonged L-DOPA treatment induces AIMs by PD model flies.**
**a** Experimental scheme to assay single fly trajectories upon acute L-DOPA diets intercalated with control diets. Thick green and gray arrows indicate time periods of L-DOPA and control diets, respectively. Black arrows indicate video recordings of the fly movement. **b** Representative trajectories of a control ($PINK1^{RV}$) and a PD model fly ($PINK1^{B9}$) upon acute L-DOPA diets. Abnormal locomotor patterns are highlighted in red circles. **c**, **d** Comparisons of the quantified speed (**c**) and AIM scores (**d**) of the control and PD model flies upon acute L-DOPA diets. $N = 6$. ****$p < 0.0001$; ***$p < 0.001$; n.s., not significant ($p > 0.05$) by two-way ANOVA Tukey's multiple comparison test and are presented as means ± SEM. $p < 0.05$ was considered statistically significant. **e** Experimental scheme to assay single fly trajectories upon chronic L-DOPA diets intercalated with control diets. Thick green and gray arrows indicate time periods of L-DOPA and control diets, respectively. Black arrows indicate video recordings of the fly movement. **f** Representative trajectories of a $PINK1^{RV}$ and a $PINK1^{B9}$ fly upon chronic L-DOPA diet. Abnormal locomotor patterns are highlighted in red circles. **g**, **h** Comparisons of the quantified speed (**g**) and AIM scores (**h**) of the control and PD model flies upon chronic L-DOPA diet. $N = 6$. ****$p < 0.0001$; **$p < 0.01$; n.s., not significant ($p > 0.05$) by two-way ANOVA Tukey's multiple comparison test. Data are presented as means ± SEM. $p < 0.05$ was considered statistically significant.

(Fig. 3d). However, $PINK1^{RV}$ flies showed an increase of AIM scores in addition to alterations in speed, freezing and yawing, indicative of LID pathogenesis (Fig. 3b–d, Supplementary Fig. 2a, b). Upon removal of L-DOPA diet on day 2, these LID-like symptoms in $PINK1^{RV}$ flies were reversed to normal state, whereas the PD-associated motor defects were again shown in $PINK1^{B9}$ flies (Fig. 3b, c, Supplementary Fig. 2a, b). Reversibly, on day 3 with L-DOPA diet, motor defects of $PINK1^{B9}$ flies were normalized, but $PINK1^{RV}$ flies exhibited LID-like phenotypes (Fig. 3b–d, and Supplementary Fig. 2).

Since PD patients develop LID after several years of L-DOPA intake, we wondered whether a prolonged L-DOPA diet by PD model flies would lead to LID. To address this possibility, we designed a seven-day L-DOPA diet experiment in which flies showed maximal AIM scores (Fig. 2f). In the experiment, a fly was provided with control diet on day 0 followed by L-DOPA diet for seven days (Fig. 3e). As shown above, PD-associated motor defects of $PINK1^{B9}$ flies on day 0 were normalized, while control flies developed LID-like phenotypes with an increase in AIM scores upon L-DOPA diet on day 1 (Fig. 3f–h, Supplementary Fig. 3a, b). After seven days of L-DOPA diet, $PINK1^{B9}$ flies also developed LID-like phenotypes with pathological increase in AIM

scores (Fig. 3h). In control flies, AIM scores were much exacerbated upon seven days of L-DOPA diet (Fig. 3f–h, Supplementary Fig. 3). Together, these data show that prolonged L-DOPA intake causes LID-like symptoms in PD model flies.

**Neuronal Dop1R1 is necessary and sufficient for LID-like symptoms.** Dopamine receptors have been implicated in LID pathogenesis by abnormally boosting the postsynaptic signaling[36]. Therefore, we asked whether *Drosophila* dopamine receptors play a role in LID pathogenesis by manipulating their expression. Like mammals, flies possess D1- and D2-like family receptors, termed Dop1R1, Dop1R2 (lacking human homolog), and Dop2R[37]. We decided to use *n-Synaptobrevin-GAL4* (*nSyb-GAL4*) to drive the expression of *UAS-Dop1R1 RNAi* and *UAS-Dop2R RNAi* in synaptically mature neurons. Neuronal knockdown of Dop1R1 expression not only reversed L-DOPA-induced motor defects to normal state, but also decreased AIM scores in both acute and prolonged L-DOPA diet (Fig. 4a–c). It is worth to note that neuronal knockdown of *Dop1R1* elevated baseline movements in control and L-DOPA diet (Fig. 4a, b). By contrast, neuronal overexpression of *Dop1R1* using a *UAS-Dop1R1*

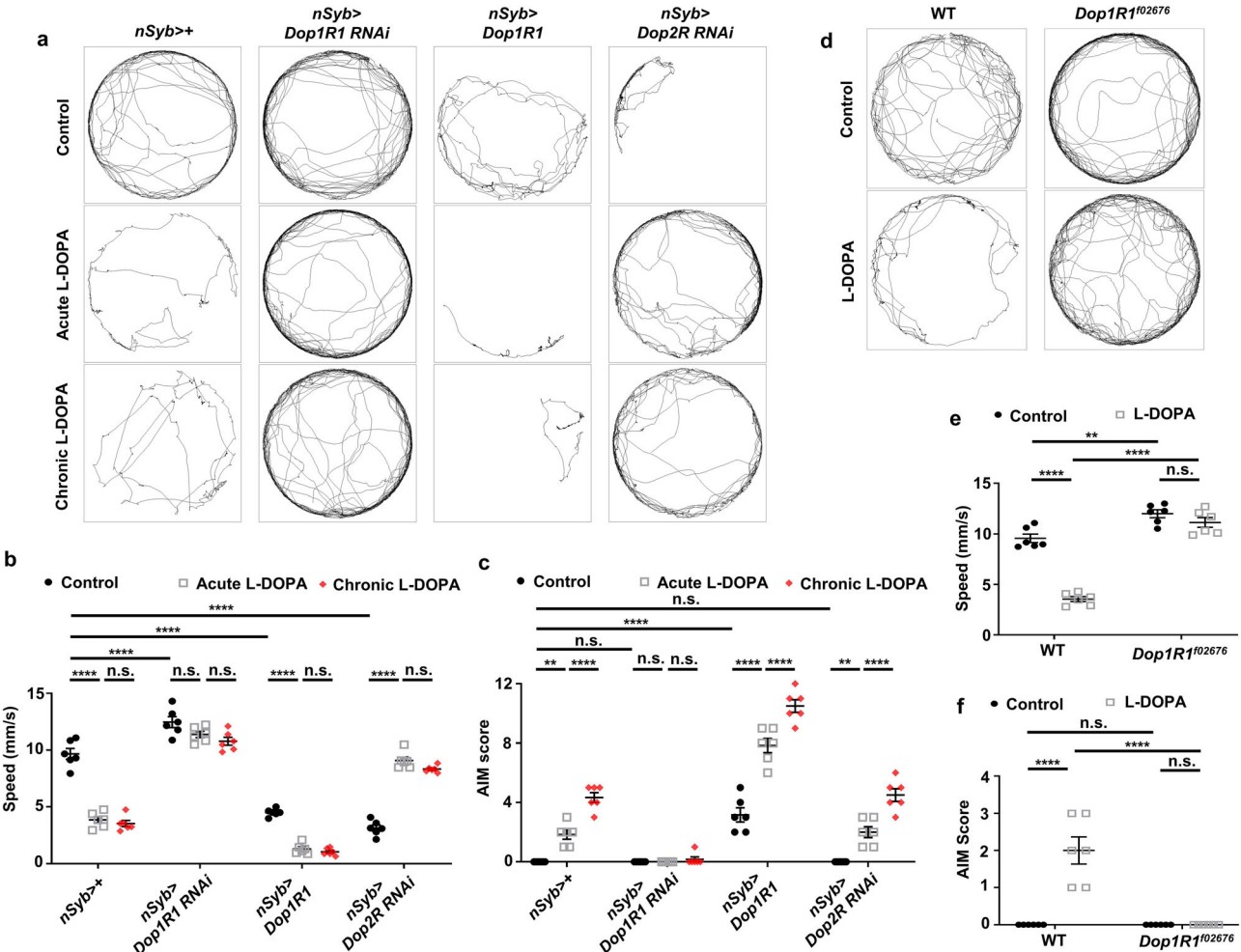

**Fig. 4 Dop1R1 is necessary and sufficient for LID-like phenotypes. a** Representative trajectories of the flies with indicated genotypes on a control, acute L-DOPA, and chronic L-DOPA diet. **b, c** Comparisons of the quantified speed (**b**), and AIM scores (**c**) of the flies with indicated genotypes. $N = 6$. ****$p < 0.0001$; **$p < 0.01$; n.s., not significant ($p > 0.05$) by two-way ANOVA Tukey's multiple comparison test and are presented as means ± SEM. $p < 0.05$ was considered statistically significant. **d** Representative trajectories of the flies with indicated genotypes. **e, f** Comparisons of the quantified speed (**e**) and AIM scores (**f**) of the flies with indicated genotypes. $N = 6$. ****$p < 0.0001$; **$p < 0.01$; n.s., not significant ($p > 0.05$) by two-way ANOVA Tukey's multiple comparison test. Data are presented as means ± SEM. $p < 0.05$ was considered statistically significant.

transgene driven by *nSyb-GAL4* markedly decreased baseline movements (Fig. 4a, b). Moreover, Dop1R1 overexpression alone increased AIM scores without L-DOPA diet (Fig. 4c), further deteriorating LID-like symptoms upon L-DOPA diet (Fig. 4a–c). On the other hand, neuronal knockdown of *Dop2R* expression decreased baseline movements on the control diet (Fig. 4a, b). Interestingly, neuronal *Dop2R* knockdown had minimal impact on AIM scores upon L-DOPA diet, while it clearly restored the baseline movement of the L-DOPA-fed flies (Fig. 4a, b).

Due to deficits in reproduction and life span of the PD model flies along with technical challenges putting multiple genetic elements together in the *PINK1* knockout background[35], we could not perform the experiment using the currently available techniques. However, it would be highly relevant to examine the effect of these dopamine receptor manipulations in the PD model flies upon L-DOPA diet conditions in parallel with the manipulation in WT flies.

To further validate the requirement of *Dop1R1* in the LID-like phenotype, we obtained *Dop1R1^f02676* fly line, previously reported as a strong hypomorphic mutant allele for *Dop1R1*[38], and asked whether *Dop1R1^f02676* flies show altered L-DOPA responses. *Dop1R1^f02676* flies showed increases of baseline movements in line

with the knockdown results (Fig. 4d, e). Importantly, *Dop1R1^f02676* flies showed marked suppression of AIM scores and improved locomotor phenotypes upon L-DOPA diet (Fig. 4d–f). Taken together, our data suggest that *Dop1R1* is necessary and sufficient for LID-like symptoms via its inhibitory role in motor control.

**GWAS on LID-diagnosed PD patients identifies genes related to LID pathogenesis.** To identify the genetic factor that makes PD patients susceptible to LID, we performed GWAS on the patients clinically diagnosed with LID (see Methods)[39]. Filtered through quality control of the chip dataset and clinical symptoms along with patient criteria involving disease duration (>5 years) and the age of onset (>50 years), five hundred seventy-eight patients were selected for GWAS (Supplementary Fig. 4). From these patients, we initially obtained 579,399 SNPs (Supplementary Fig. 4). Further statistical validations ($p < 0.000001$), and sorting of SNPs within protein-coding genes and open reading frames reduced the number of SNPs down to one hundred seventy four. These SNPs were then analyzed for their genetic loci on the human genome, leading to identification of one hundred

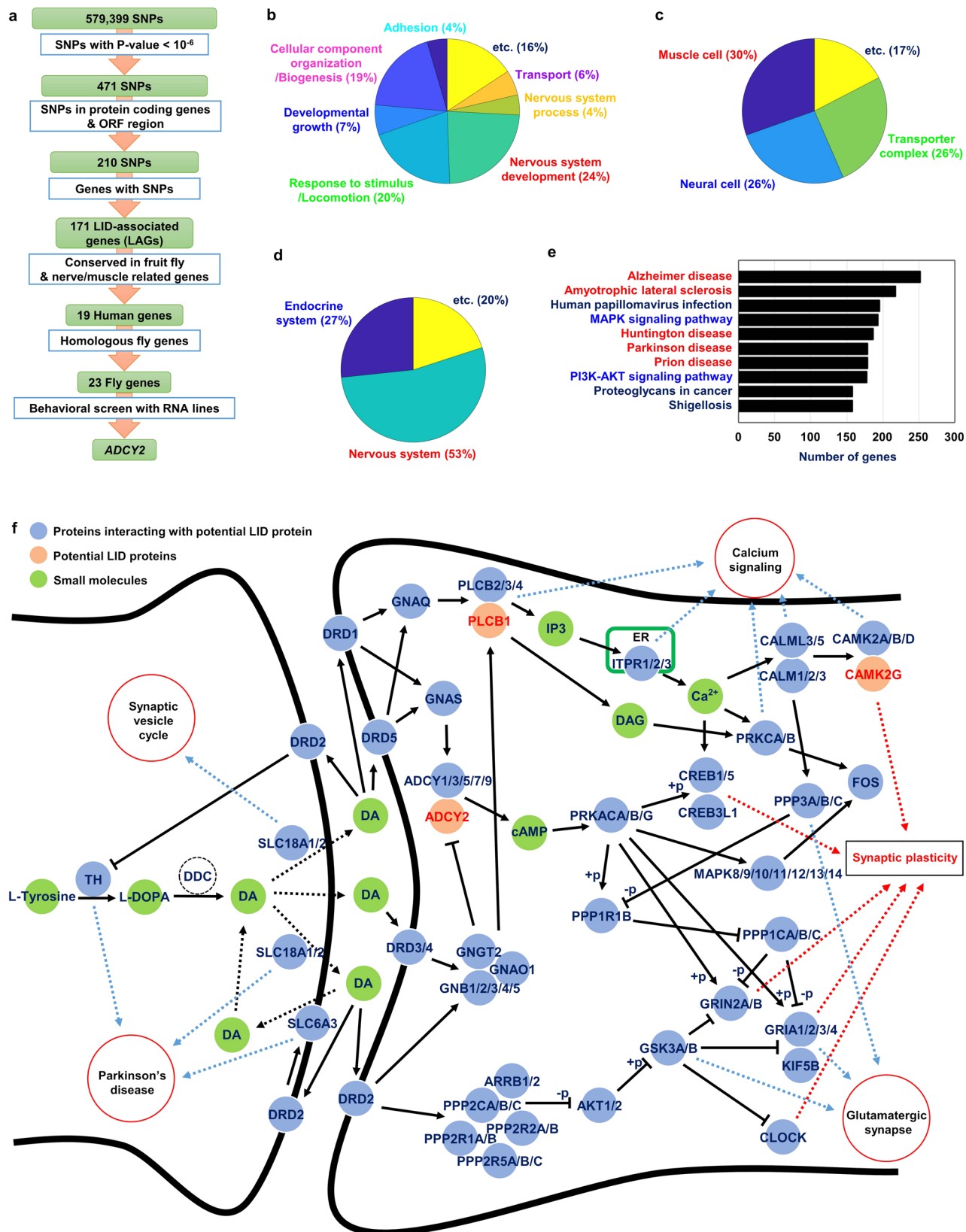

seventy-one genes potentially correlated with LID (Fig. 5a, 1st–4th rows). To establish functional categories of these LID-associated genes (LAGs), we conducted Gene Ontology (GO) enrichment analyses. From enriched analyses of biological processes, we discovered that LAGs were enriched in the category of nervous system development, response to stimulus/locomotion,

and cellular component organization/biogenesis (Fig. 5b, Supplementary Fig. 5a). With cellular component analyses, LAGs were enriched in the category of muscle cell, neural cell and transporter complex (Fig. 5c, Supplementary Fig. 5b). To gain further functional insights into LAGs, we analyzed their related pathways using KEGG pathway analysis. Most of the genes were

**Fig. 5 dADCY2 functions downstream of Dop1R1 for LID-related phenotypes. a** A flowchart illustrating GWAS and screening strategies. **b, c** GO enrichment analysis of LAGs. Pie charts represent the enriched GO annotation categories of biological process (**b**) and cellular component (**c**). **d** Pathway analysis of LAGs. Pie charts represent the mapped pathway categories of KEGG pathway. **e** Pathway analysis on genes comprising LAPN. Reds indicate neurodegenerative disease pathways. Blues indicate signal transduction pathway. **f** Dopaminergic synapse pathway map constructed by extracting dopaminergic synapse sub network from LRPN with their interacting molecules. Solid orange nodes indicate potential LID proteins encoded by LAGs. Solid light blue nodes indicate proteins interacting with the potential LID protein that serves as a neighboring node in LAPN. Solid light green nodes indicate small molecules. Red lined circles indicate pathways that interact with the LID-associated dopaminergic synapse pathway.

enriched in nervous and endocrine system categories, suggesting LAGs are involved in signaling in the nervous system (Fig. 5d, Supplementary Fig. 5c, and Supplementary Table 1). For a topological map of LAGs, we sought to perform a series of network analyses. By integrating major databases of gene interaction and protein-protein interaction (PPI), we built the integrated human PPI functional network (Supplementary Fig. 6, and see Methods). From this original network, we extracted LAGs and their neighboring genes to create a LID-associated PPI network (LAPN) that harbors 6483 genes (nodes) and 9718 interactions (edges) (Supplementary Fig. 6). Based on the enriched genes from KEGG pathway analyses on LAPN, we constructed LID-related pathway network (LRPN) as a subnetwork that possesses reduced number of nodes and edges down to 1255 and 1524, respectively (Supplementary Figs. 6 and 7). In this pathway analysis, neurodegenerative disease and the disease-related signal transduction pathways were highly ranked (Fig. 5e). To assess the density of LAPN, we examined another subnetwork in which terminal nodes were eliminated. This subnetwork possesses 2245 nodes and 5481 edges that count 34.64% and 56.4% less than those within LAPN (Supplementary Figs. 6 and 8). The percent reduction in edges greater than that of nodes suggests that the density of LAPN is solid (Supplementary Fig. 8). Next, we wondered whether LAGs are indeed correlated with dopamine signaling. By extracting dopaminergic synapse subnetwork from LRPN, we visualized a dopaminergic synapse pathway map (Fig. 5f, Supplementary Fig. 6). LAGs involving *ADCY2*, *phospholipase C beta-1* (*PLCB1*), and *calcium/calmodulin-dependent protein kinase II gamma* (*CAMK2G*) were mapped downstream of the dopamine receptors. In particular, ADCY2 and PLCB1 were direct targets of D1 type receptors depending on the types of coupling with G-proteins. Dopaminergic synapse pathway predicts various pathogenic routes via the postsynaptic signaling system including synaptic vesicle cycle, calcium signaling, glutamatergic synapse pathway (Fig. 5f)[40–42]. Therefore, it is conceivable that dysregulation of genes in the dopaminergic synapse pathway can result in altered synaptic plasticity that would cause LID (Fig. 5f).

**dADCY2 functions downstream of Dop1R1 for LID-related phenotypes**. Based on conservation of LAGs in the *Drosophila* genome and association with nerve and muscle, nineteen genes were chosen for test in the fly LID behaviors (Fig. 5a, 4th and 5th rows). To test these genes in fly locomotion, we obtained twenty-three RNAi lines against those genes without known off-targets (Fig. 5a, 6th row, Supplementary Table 2). Using these RNAi lines driven by *nSyb-GAL4*, we searched for candidates that alter AIM scores and locomotion. We noticed that neuronal knockdown of *adenylyl cyclase 76E* (*Ac76E*; *dADCY2* based on its homology to human *ADCY2*) increased baseline movements both in control and L-DOPA diet, and strongly suppressed the pathological AIM increase (Fig. 6a, b, #22). This was a phenotypically reminiscence of *Dop1R1* knockdown and knockout result, and ADCYs are the major downstream target of dopamine receptors[43]. These prompted us to examine the possibility that dADCY2 functions downstream of Dop1R1. Remarkably, neuronal knockdown of

*dADCY2* blocked both the Dop1R1 overexpression-induced decrease of baseline movements and AIM increase (Fig. 6c–e). These data suggest that dADCY2 is a critical downstream target of Dop1R1 in LID pathogenesis, supporting the notion that dopaminergic synapse pathway map predicted ADCY2 functions downstream of D1 receptor (Fig. 5f).

**Phenotypical classification of ADCY family reveals four distinct groups**. Multiple types of ADCYs are expressed in the dorsal striatum, the brain region associated with LID[44]. Diverse signals downstream of dopamine receptors converge on ADCYs to regulate cAMP production and this enables the web of downstream signaling in the striatal neurons[43]. However, implication of other members of ADCY family in L-DOPA-induced pathologies has not been fully addressed. According to our alignment analyses, there are at least six conserved *Drosophila* ADCY genes, including *rut* (*dADCY1*), *dADCY2*, *Ac3* (*dADCY3*), *CG43373* (*dADCY5*), *Ac78C* (*dADCY8*), and *Ac13E* (*dADCY9*) (Supplementary Table 3). Our phylogenetic analyses revealed that human ADCY genes are classified into four groups composed of *ADCY1/5/6*, *ADCY2/4/7*, *ADCY3/8*, and *ADCY9/10* (Supplementary Fig. 9a, c). Interestingly, the fly ADCY genes were also categorized into four groups, *dADCY1/5*, *dADCY2*, *dADCY3/8*, and *dADCY9* consistent with human gene classification, suggesting that each group may play evolutionarily distinct functions (Supplementary Fig. 9b, c). We categorized *dADCY1/5*, *dADCY2*, *dADCY3/8*, and *dADCY9* into Group1, Group2, Group3, and Group4, respectively, based on their homology to human ADCYs (Fig. 7a, Supplementary Table 3). We sought to knock down each of the ADCY members in the fly nervous system and examined their effects on the baseline movement and LID-like symptoms by measuring speed and AIM scores. For reliable RNAi knockdown, we used ADCY RNAi lines that have been validated by others[44–47]. Neuronal RNAi knockdown of *dADCY1/5* normalized the baseline movement and LID-like symptoms upon acute L-DOPA diet. By contrast, it had no impact on the baseline movement, while it caused LID-like symptoms upon control and chronic L-DOPA diet (Fig. 7a–e), indicating that dADCY1/5 mediate the early state of LID-like symptoms without affecting the baseline movement in control diet. As described above, neuronal knockdown of *dADCY2* completely suppressed LID-like symptoms upon acute and chronic L-DOPA diet, while it increased the baseline movement upon all three diets (Fig. 7a–e). These results suggest that ADCY2 is involved in early and late phase of LID-like symptoms via inhibitory motor regulation. Neuronal knockdown of *dADCY3/8* markedly decreased the baseline movement, but it suppressed LID-like symptoms upon acute L-DOPA diet. Interestingly, neuronal knockdown of *dADCY3/8* rescued the baseline movement, but not LID-like symptoms upon chronic L-DOPA diet (Fig. 7a–e), suggesting that dADCY3/8 mediates early state of LID-like symptoms by positively regulating motor control. Neuronal knockdown of *dADCY9* suppressed LID-like symptoms, whereas it increased the baseline movement upon acute and prolonged L-DOPA diet. Interestingly, however, it had no impact on the baseline movement on control diet (Fig. 7a–e). This is the major difference between *ADCY2* and

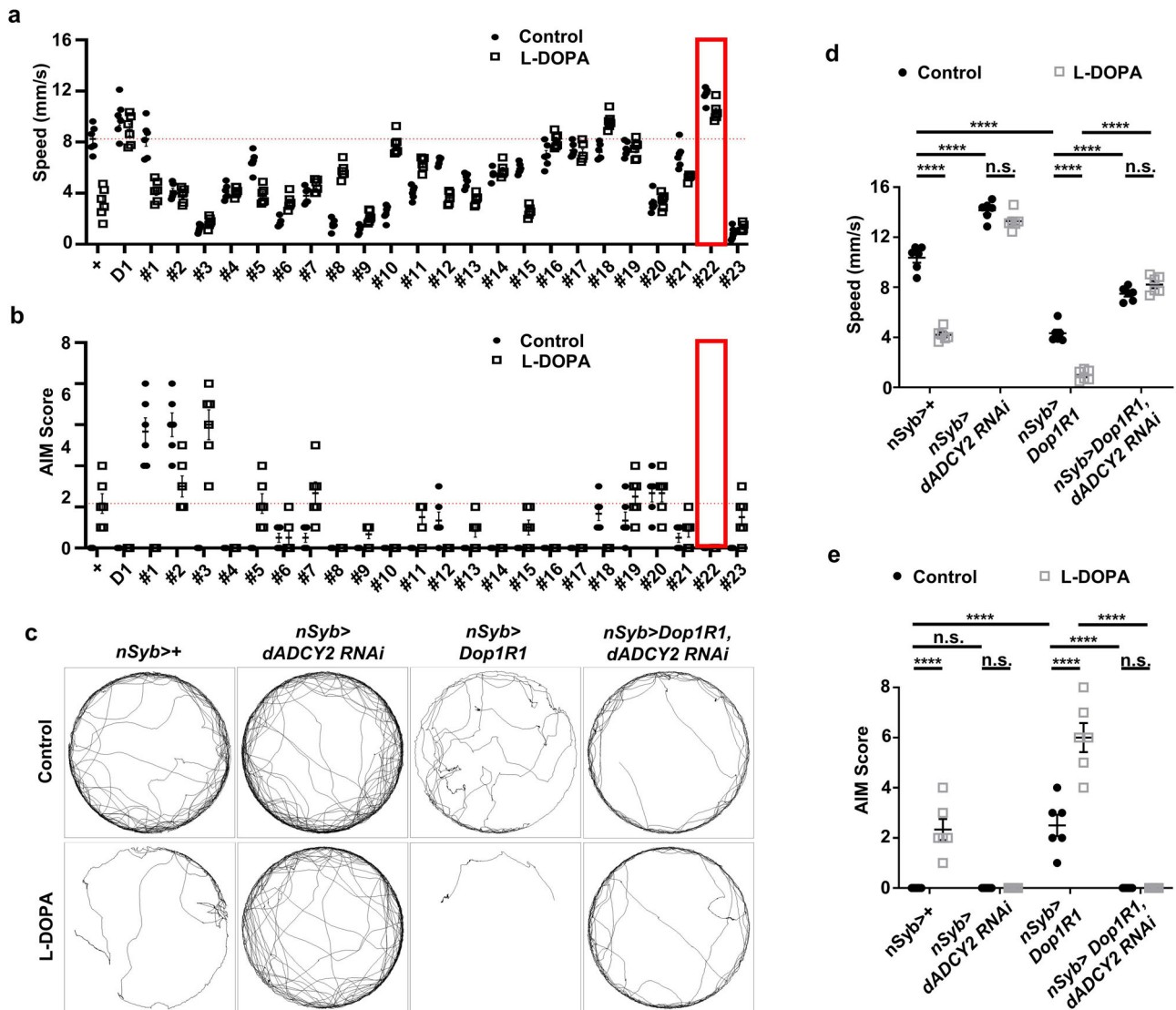

**Fig. 6 dADCY2 functions downstream of Dop1R1 for LID-related phenotypes. a, b** Comparisons of the quantified speed (**a**) and AIM scores (**b**) of the flies in which twenty-three LID-related genes were pan-neuronally knocked down by RNAi expression driven by *nSyb-GAL4*. + denotes the flies bearing a copy of *nSyb-GAL4* driver with no RNAi expression. D1 indicates the flies expressing Dop1R1 RNAi driven by *nSyb-GAL4*. Red boxes indicate *Drosophila ADCY2* RNAi driven by *nSyb-GAL4*. Black (control) and white bars (L-DOPA) indicate control and L-DOPA diet. The mean speed and AIM score are depicted in red line. $N = 6$. **c** Representative trajectories of the flies with indicated genotypes upon control and L-DOPA diet. **d, e** Comparisons of the quantified speed (**d**) and AIM scores (**e**) of the flies with indicated genotypes. $N = 6$. ****$p < 0.0001$; n.s., not significant ($p > 0.05$) by two-way ANOVA Tukey's multiple comparison test. Data are presented as means ± SEM. $p < 0.05$ was considered statistically significant.

*ADCY9* knockdown phenotypes as *ADCY2* knockdown increases the baseline movement phenocopying *Dop1R1* knockdown. We propose that ADCY2 and ADCY9 may share a common cellular and molecular pathway in the pathogenesis of LID symptoms, while the role of ADCY9 in the baseline movement is modulatory (dispensable) and independent of Dop1R1. Together, we identified four functionally distinct groups of ADCYs in response to L-DOPA resembling the phylogenetic categorization (Fig. 7d–f).

## Discussion
In the current study, we have shown that L-DOPA intake produces prominent effects on fly locomotion highly correlated with its intensity and duration. These effects include decrease of mean speed, increased freezing, and yawing along with abrupt peaks of instantaneous speeds (hyperkinetic movements). We have demonstrated that the hyperkinetic movements were deviated from

the regular movements at the near time period and were manifested in an uncontrolled fashion (involuntary), reminiscent of AIMs in LID. To model fly AIMs, we developed a computational algorithm that enables systematic quantification of AIM scores from fly trajectories. Remarkably, prolonged L-DOPA diet induced the pathological increase of AIM scores in PD model flies, while acute L-DOPA diet improved the motor abnormalities of PD model flies with no sign of AIMs. We found that either neuronal knockdown or whole-body knockout of *Dop1R1* expression strongly suppressed LID-like phenotypes, whereas neuronal overexpression of *Dop1R1* was sufficient to evoke LID-like defects even without L-DOPA treatment. Using a collection of SNPs from human PD patients susceptible to LID, we obtained nineteen candidate genes that are conserved in the fly. We performed a neuronal RNAi knockdown screen and identified *dADCY2*, which resulted in a complete suppression of LID-related defects. Neuronal RNAi knockdown of *dADCY2* also suppressed the LID-associated defects caused by

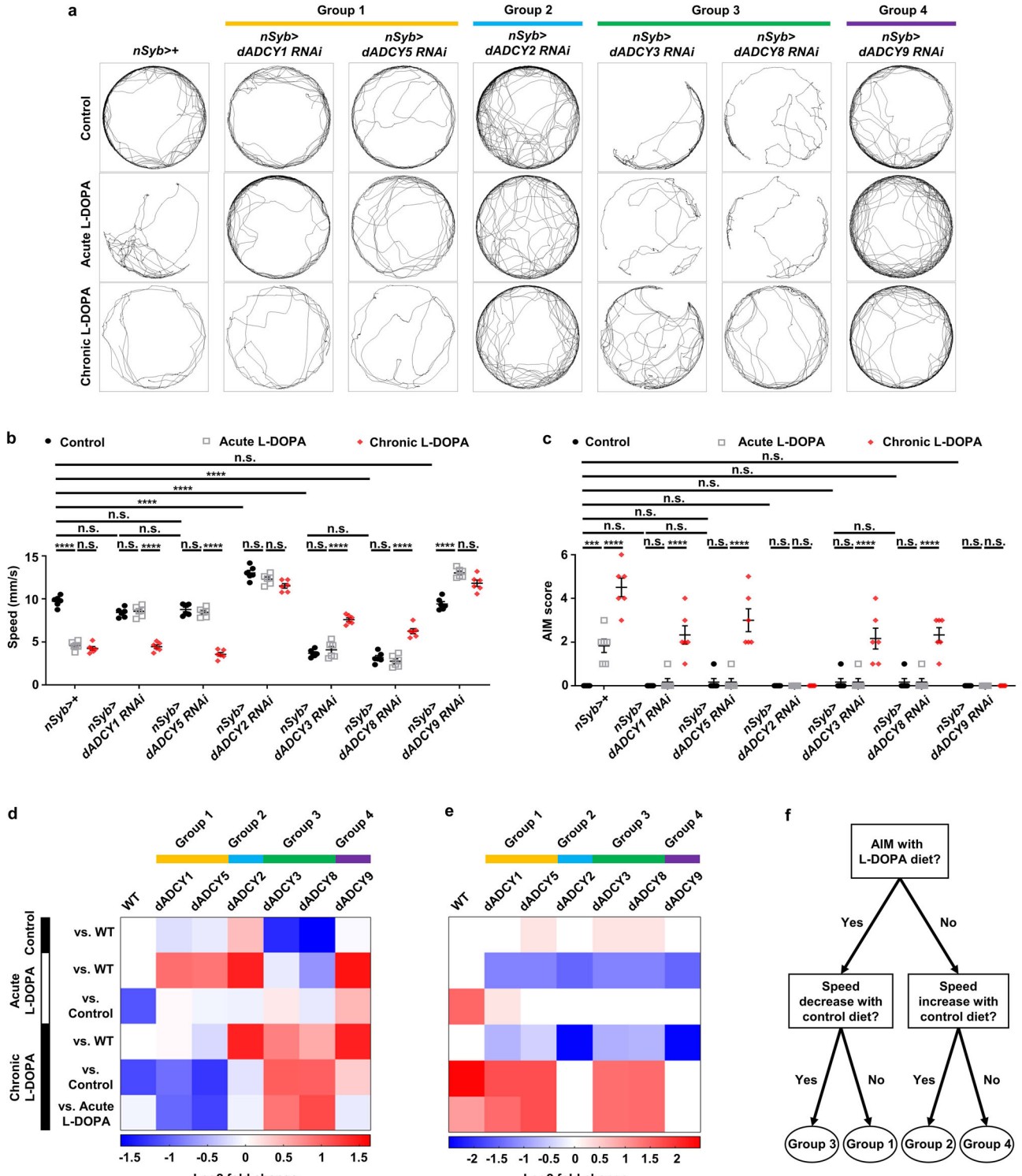

**Fig. 7 Phenotypical classification of the fly ADCY family reveals four distinct groups. a** Representative trajectories of the flies with indicated genotypes upon control, acute and chronic L-DOPA diet. **b**, **c** Comparisons of the quantified speed (**b**) and AIM score (**c**) of the flies with indicated genotypes upon control, acute, and chronic L-DOPA diet. $N = 6$. ****$p < 0.0001$; ***$p < 0.001$; n.s., not significant ($p > 0.05$) by two-way ANOVA Tukey's multiple comparison test. Data are presented as means ± SEM. $p < 0.05$ was considered statistically significant. **d**, **e** Heatmaps of quantified speeds and AIM scores in the flies with pan-neuronal knockdown of the four ADCY groups upon normal diet (control), acute and chronic L-DOPA diet. Blues indicate decrease and reds indicate increase of the log2 fold change. **f** A decision tree to illustrate the selection criteria for the ADCY groups.

overexpression of *Dop1R1*, suggesting that dADCY2 functions downstream of Dop1R1.

Taken together, we propose that the D1-like family receptor-ADCY2 signaling axis in the postsynaptic dopamine neurons

slows down locomotor speed in normal situations. Upon L-DOPA diet, non-physiologically high levels of dopamine are synthesized by L-DOPA-dopamine conversion pathway in the presynaptic neurons. This can lead to pathogenic levels of D1-like

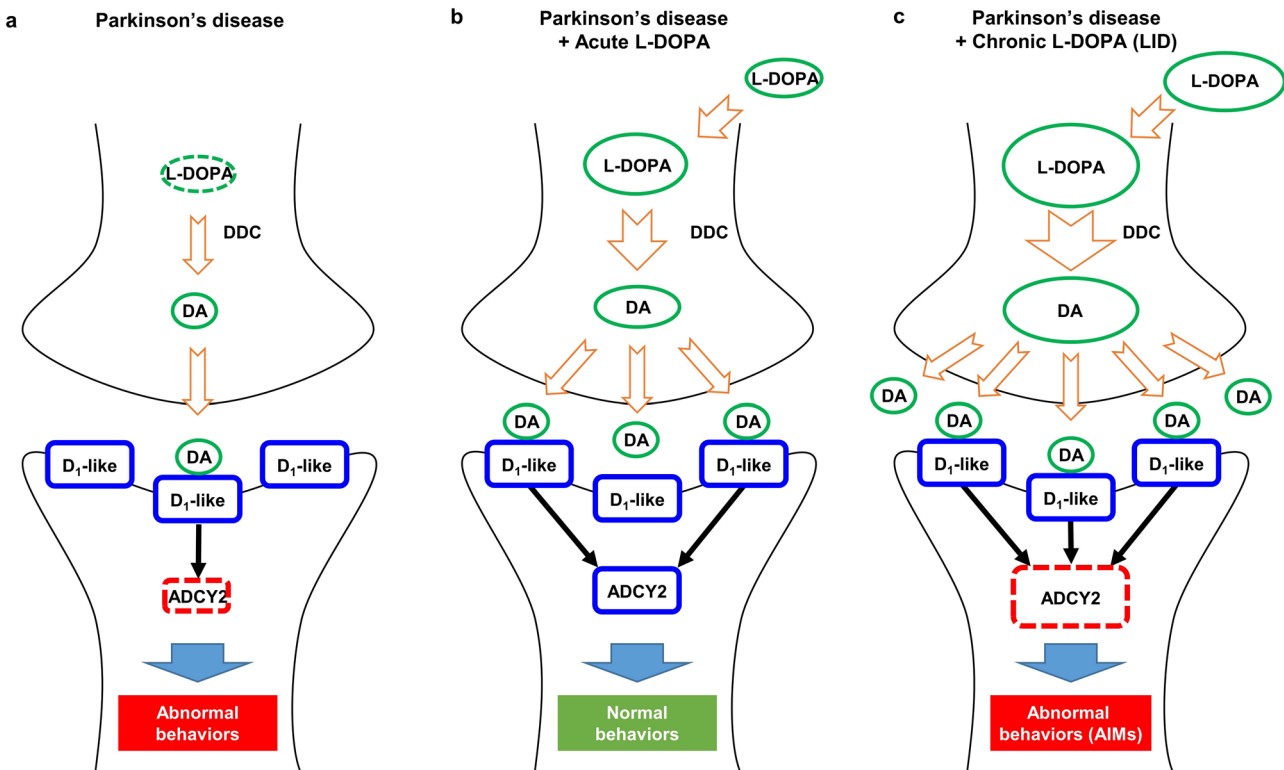

**Fig. 8 A proposed mechanistic model of LID in *Drosophila*. a** Schematic illustration of PD pathogenesis. Dampened DA production in the presynaptic terminals leads to depression of postsynaptic D1-like receptor-ADCY2 signaling causing motor deficits. **b** Schematic illustration of acute L-DOPA-mediated amelioration of PD. Acute L-DOPA administration normalizes presynaptic DA production and the postsynaptic signaling is restored, resulting in normal motor behaviors. **c** Schematic illustration of LID pathogenesis. Chronic L-DOPA administration causes excessive DA conversion in the presynaptic terminals and the excessive DA abnormally boosts of postsynaptic D1-like receptor-ADCY2 signaling, leading to LID. DA, dopamine; D1-like, D1-like family receptors; DDC, DOPA decarboxylase.

family receptor stimulation on the postsynaptic terminals, which in turn hyper-stimulates ADCY2. Hyper-stimulation of ADCY2 results in exaggerated signaling in the web of downstream molecules and eventually, it alters the function and structure of the postsynaptic neurons to cause LID. In the model, we propose that dysregulation of the D1-like family receptor-ADCY2 signaling axis underlies LID pathogenesis (Fig. 8).

LID is classically studied using rodents and primates. Recently, however, a study has reported the characterization of L-DOPA-induced motor abnormalities in a simpler model organism such as the *Drosophila* larvae, providing another useful model system to study LID[32]. In the study, the authors generated a PD model of *Drosophila* larvae expressing human mutant α-synuclein (A53T) and characterized a body bending behavior upon L-DOPA feeding, which corresponds to axial dyskinesia in the rodent model of LID. In our study, we identify conserved *Drosophila* genes pathogenic for LID using combination of GWAS on LID-diagnosed PD patients, and genetic studies on the adult *Drosophila* model of LID. Our unbiased searches of natural variant mutations on the patient genome allowed us to identify LID-associated genes (LAGs). Then, we crafted a computational analysis of the *Drosophila* locomotion to capture motor traits that resemble the symptoms of the rodent model of LID including the AIMs. By examining the conserved LAGs in the fly LID paradigm, we identified *ADCY2*, whose role in LID has not been previously reported. The approaches used in this study offer three main implications for the field. Our *Drosophila* model of LID provides a quick and reliable in vivo platform to evaluate the LID motor symptoms from various angles by using fluent genetic tools to manipulate each of the molecular and cellular components of

the dopaminergic system. Our GWAS establishes a patient genomic dataset that lists natural variants potentially associated with LID, allowing us to investigate additional molecules involved in the pathogenesis of LID. Our collaborative study establishes a framework for the combined research on LID pathogenesis between clinical data and mechanistic studies using animal models.

Our study raises some open questions. What is the pathogenic mechanism underlying the occurrence of LID by the SNP on *ADCY2* in PD patients? It is conceivable that the mutation on *ADCY2* induces pathological changes in ADCY2 expression/activity in the patient brain. For example, a gain-of function mutation on ADCY5 with increased activity was shown to correlate with familial dyskinesia[48]. Intriguingly, studies show that striatal ADCY5 levels were elevated upon L-DOPA treatment and knocking out of ADCY5 ameliorated L-DOPA-induced pathological symptoms in the PD model mice[24]. Alternatively, in our study, the *ADCY2* mutations could induce pathogenic effects above the threshold set by pulsatile dopamine fluctuations during the course of L-DOPA treatment. In this case, we may need to monitor changes in ADCY2 expression/activity before and after L-DOPA therapy to capture the pathological changes. It would be difficult to address all these questions directly in humans, but one way to test could be to design a mutation mimicking the SNP on the *ADCY2* locus in animal models. In vivo and in vitro examination of the expression/activity of the mutant ADCY2 could provide mechanistic insight into how the SNP on *ADCY2* influences LID pathogenesis.

A portion of *PINK[B9]* flies with the mild AIM score upon the prior L-DOPA diet displayed a somewhat reversed phenotype

during the non-L-DOPA diet (Fig. 3). We want to ensure that the AIM score is too low to be justified for a statistical significance. The effect of L-DOPA early in the course of the treatment is usually beneficial and long-lasting with reduced likelihood of dyskinesia, and the dyskinetic symptoms may be mild in the early phase of LID[49,50]. Given that the acute L-DOPA treatment is largely beneficial to the PD symptoms (speed and trajectory) of the mutant fly, reversibility here may represent the therapeutic effect during the early treatment, and the mild AIM score from the L-DOPA treatment may indicate a very early phase of LID progression. It is generally accepted that once LID develops, it is difficult to reverse the process. Low dosing of L-DOPA can be an option, but the dyskinetic symptoms may still persist. Pulsatile stimulation of the receptors in the dopaminergic pathway is thought to underlie the abrupt appearance of dyskinetic symptoms[6]. This offers a clinical opportunity that L-DOPA dosing complemented with pharmacological intervention of the receptors underlying the pulsatility may be a way to reverse the symptoms. Interestingly, a recent study reported that viral delivery-guided inhibition of CaV1.3 channels in the striatum reversed the dyskinetic symptoms of animal models with severe LID. Importantly, the motor benefit from low dosing of L-DOPA was still preserved with the inhibition of the CaV1.3 channels, suggesting that this ion channel may underlie the reversibility of LID pathogenesis, independently of the dyskinesia-inducing mechanism[51].

Our data show that each dADCY group is involved in distinct motor controls at multiple time points during L-DOPA treatment. Perhaps, ADCYs engage different cellular signaling machineries in a discrete neuronal subset in response to L-DOPA[52]. Depending on which upstream receptors and downstream signaling components are coupled, how strong the signaling cascade persists, and which cells they are expressed in, there could be temporal and spatial dimensions in which these ADCYs interplay in mediating LID pathogenesis. Mammalian ADCY5 is the best studied ADCY member for its role in a variety of tissues and physiologies[53,54]. ADCY5 is implicated in a protective role in the heart, including responding to pressure overload and cardiac stress through the ADCY signaling in ~~the~~ cardiomyocytes[54]. ADCY5 knockout increases longevity in mice via activation of Raf/MEK/ERK signaling pathway and upregulation of protective molecules in fibrocytes[53]. ADCY5 in the striatum neurons responds to L-DOPA treatment and has a protective role in the pathogenesis of LID in mice when knocked out[24,55,56]. ADCY3 mediates the rhythmic control of the morning anticipatory response by functioning in a subset of clock cells in the *Drosophila* brain[45]. Likewise, ADCYs mediate diverse functions in tissues with the ability to time their functions. Using a genetically tractable model organism such as *Drosophila* which permits fluent cell-specific manipulations and recordings of ADCYs, it would be possible to further address the temporal and spatial pattern of ADCYs expression in LID-associated tissues and their function in the L-DOPA-induced motor response.

It is noteworthy that we found additional genes that suppressed the pathological increase of L-DOPA-induced AIMs. We could not rule out the possibility that RNAi knockdown of these genes merely produced unrelated locomotor defects. Interestingly, however, some of these genes were picked up multiple times and are involved in the regulation of neuronal excitability, cellular signaling, and protein synthesis in neurons. In particular, it is intriguing to find that neuronal RNAi knockdown of two fly genes, *S6KII* and *S6KL*, orthologous to human ribosomal protein S6 kinase alpha-2 (*RPS6KA2*) and downstream members of ERK-MAP kinase and mTOR signaling pathways, effectively normalized L-DOPA-induced AIMs (Fig. 6a, b, Supplementary Table 2). Studies have postulated that mTOR and ERK-MAP kinase

signaling play an important role in LID pathogenesis via dysregulation of postsynaptic protein synthesis and gene expression ultimately causing pathogenic changes in dopaminergic signaling[57,58]. In further support of the view, various SNPs found on mTOR and ERK-MAP kinase signaling-related genes appeared to be associated with LID susceptibility in PD patients[59]. Taken together, our study will provide additional insights into the mechanism underlying LID pathogenesis through further validation of mTOR signaling-related genes.

## Methods

**Fly stocks and husbandry**. All the fly stocks were reared on a regular fly food provided from Korea Advanced Institute of Science and Technology, and maintained with light/dark (12/12 h) cycle at 25 °C with 70% humidity. *Drosophila* RNAi stocks were purchased from the Bloomington Drosophila Stock Center (BDSC), and the Vienna Drosophila Resource Center (VDRC) (Supplementary Table 2). *dADCY1* RNAi, *dADCY3* RNAi, *dADCY8* RNAi, *Dop1R1* RNAi, *dADCY9* RNAi, *dADCY5* RNAi, *Dop2R* RNAi, and *nSyb-GAL4* were from BDSC. *dADCY2* RNAi line was from VDRC. *Dop1R1*[f02676] line was obtained from Harvard Medical School Exelixis Collection. *PINK1*[B9] or *PINK1*[RV] lines used in our study were previously generated by imprecise or precise excision of a P element inserted in the *Drosophila PINK1* locus[35]. In the experiment for Fig. 6, the number of UAS elements was not matched.

**Modeling of fly AIM scores**. The instantaneous speed ($v_i$) equals the subtracted value from the traveled distance at the next time point ($d_{i+1}$) to the traveled distance at a time point ($d_i$).

$$v_i = d_{i+1} - d_i \qquad (1)$$

The mean speed ($v_s$) was obtained from a window size. The interval ($s$) was 1/2 ($s$-1), the region before and after a time point ($i$), and s was defined as the window size. The sliding window was scanned over the entire frame to calculate the serial ratio ($v_i/v_s$) within it[60].

$$v_s = \frac{1}{s}\left(\sum_{j=i-[\frac{s}{2}]}^{i-1} v_j + \sum_{k=i}^{i+[\frac{s}{2}]} v_k\right), s = \text{window size} \qquad (2)$$

The logarithmic value of $v_i/v_s$ (log $v_i/v_s$) was then compared to a threshold value ($c$). $c$ was experimentally determined from WT fly movements on control diet. For better visualization of data, the AIM score of control flies is normalized to be 0, where the threshold value ($c$) is 0.4.

$$H = \sum_{i=0}^{n-s}\left\{\textit{if}\left|\log_{10}\frac{v_i}{v_s}\right| > c = 1, \text{ otherwise} = 0\right\}, \mathbf{c} = \text{threshold constant} \qquad (3)$$

The AIM score ($H$) is the sum of 0 or 1 given to each frame (9,018 frames total/5 min) during the entire recording based on whether the value of log ($v_i/v_s$) at a frame qualifies the threshold value ($c$). If the value of log ($v_i/v_s$) is larger than c, 1 is given to the frame. If the value of log ($v_i/v_s$) is smaller than c, 0 is given. For example, if a fly showed the value of log ($v_i/v_s$) larger than $c$ at seven different frames, the AIM score of the fly is 7.

**Video tracking of fly movements**. Newly born flies were collected under $CO_2$ anesthesia and aged for 1-2 days. Female flies were not used due to their egg-laying behaviors. Flies were fed with a series of different concentrations of L-DOPA (Supelco, #PHR-1271), or D-DOPA (Sigma-Aldrich, #D9378) dissolved in 10% sucrose for 1 day as acute L-DOPA diet, or for 7 days as chronic L-DOPA diet. Using an oral aspirator, a single fly was carefully transferred into an arena made of a petridish (SPL, #10060) filled with silicon (Dow Corning, Sylgard 182 Silicone Elastomer kit). To ensure reliable object detection in Ctrax fly tracking program[61], two-dimensional movements of the fly were permitted by filling the arena space <2 mm below the top of the arena with silicon. An insulated recording chamber (675 mm × 440 mm × 410 mm) was built with a white LED light on the inside wall and a white table at the bottom under a camera (Logitech, HD pro-webcam C910). Flies were acclimated in the chamber for 1 h prior to testing. Exposure, gain, focus, and white balance of the camera were adjusted for optimal object detection. Total trajectories of the fly were recorded for five minutes with 30.06 frames per second. A frame corresponds to 0.033 seconds.

**Computational analyses of fly movements**. The 5 minute-video (9018 frames total) was analyzed using Ctrax program, fly movements at every 0.033 s were tracked. The values of speed, angular velocity, acceleration, distance, orientation and direction were obtained using Behavioral Microarray MATLAB toolbox available from Ctrax website. AIM score, mean speed, yawing, freezing rate and trajectory graphs were calculated by modified MATLAB codes.

**Life span assay**. Flies were reared on agar media containing 10% sucrose plus D-DOPA or L-DOPA at 25 °C. Percent of surviving flies was recorded daily when flies were transferred to fresh media.

**PPI functional network analysis**. Whole human functional PPI network was built by combining the existing databases of protein-protein interaction and gene-gene interaction. The human protein/gene interaction databases involving BioGrid, HumanNet, IntAct, MINT, BIND, and DIP were integrated[62]. Human functional PPI network composed 18,891 genes and 82,657 interactions (Supplementary Fig. 6).

**Gene ontology enrichment analysis and pathway analysis**. Gene ontology enrichment and pathway analyses were performed on LAGs using the Database for Annotation, Visualization, and Integrated Discovery (DAVID). The member of genes in the enriched pathway from pathway analysis was obtained using KEGG mapper.

**Phylogenetic analysis**. The orthologs of the ADCY family proteins were extracted from HomoloGene (NCBI) and the Pan-taxonomic compara (Ensembl genome) database. The paralog sequences for human and *Drosophila* ADCY family members were obtained from Uniprot. These sequences were then aligned and analyzed using the Clustal Omega software to generate multiple sequence alignments and construct phylogenetic trees.

**Human subjects**. 1,070 study subjects diagnosed with PD were recruited. Patients were enrolled by the clinical practice in the Department of Neurology of Asan Medical Center (Seoul, Republic of Korea) from 1 January 2011 to 30 April 2016. Institutional Review Board (IRB) of Asan Medical Center approved the study, and all subjects provided informed consents in accordance with the IRB regulations[39]. Demographic characteristics of the patient groups were investigated at each stage of genome-wide association study[39]. Men PD patients were 507 (47.4%) and women patients were 563 (52.6%). The mean age at the onset of PD was 58.7 years old. The mean of disease duration from the onset to the last follow-up was 9.1 years. The group of 741 patients with disease duration over 5 years was composed of 325 men (43.9%) and 416 women (56.1%) (Supplementary Fig. 4). In the group, the mean age at the onset of PD was 57.1-years old, and the mean disease duration from the onset of PD to the last follow-up was 10.8 years. The group of 578 patients with onset age over 50 years old consisted of 247 men (42.7%) and 331 women (57.3%) (Supplementary Fig. 4). The mean age at the onset of PD was 61.1 years old in this group, and the mean disease duration from the onset to the last follow-up was 10.3 years[39].

**GWAS using PD patients with LID**. The association of each genetic variant with the occurrence of LID was investigated using GWAS including statistical analysis[39]. A GWAS flow diagram was provided to describe the number of subjects and analyses performed (Supplementary Fig.). Genotyping data were obtained using Affymetrix Axiom KORV1.1 (4845-301, 3000-3031) to identify the loci associated with the occurrence of LID-related motor fluctuations 5 years after the onset of PD[39]. It contains ~827,400 SNPs and consists of 505k ASN-based grid, 149k cSNPs & InDels, 84k KOR-based Grid, 44k Novel Exome/LOF variants, 32k Miscellaneous Variants, 17k eQTLs Markers, and 2k variants. Procedures including patient genotyping, quality control of patient samples and SNPs, and statistical analysis for extracting LID-associated SNPs were performed[39,63].

**Statistics and reproducibility**. All behavioral experiments were independently performed and measured for statistical significance evaluation. Details of quantification and statistical validations were indicated for each experiment in Figure Legends. All P-values were evaluated by one-way or two-way ANOVA Tukey's multiple comparison test, except for the life span assay of which P-value was assessed by log-rank (Mantel-Cox) test. $p < 0.05$ was considered statistically significant. Data are presented as mean ± SEM. Statistical analyses were calculated with GraphPad Prism and R Statistics for GWAS were performed as previously stated.

**Reporting summary**. Further information on research design is available in the Nature Research Reporting Summary linked to this article.

## Data availability

The authors declare that all data supporting the findings of this study are available within the article and its supplementary information files. The source data underlying the main figures that support the findings of this study are available in Supplementary Data 1, as well as Mendeley Data with the identifier data https://doi.org/10.17632/48rhgh28xn.1. The GWAS summary and human genomic data that support the findings of this study are available in https://doi.org/10.3389/fneur.2020.00570 and dbSNP, https://www.ncbi.nlm.nih.gov/SNP/snp_viewBatch.cgi?sbid=1063124.

## Code availability

The authors declare that MATLAB code for modeling of fly AIM score supporting the findings of this study are available within the article and its supplementary information files (Supplementary Note 1).

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

## Acknowledgements

We would like to thank Bloomington Drosophila Stock Center, the Vienna Drosophila RNAi Center, Harvard Medical School and the National Institute of Genetics for fly stocks. We also thank Deborah Kang for initial English writing. This work was supported by grants from the National Research Foundation of Korea (NRF) funded by the Korean government (MSIT) (NRF-2020R1A5A1018081 and NRF-2021R1A2C1010577 to J.C., NRF-2014H1A2A1022325 Global Ph.D. Fellowship Program and NRF-2020M3H1A1073304 K-BIO KIURI Center to W.Y., and NRF-2012 Global Ph.D. Fellowship Program to S.M.). J.C. and S.J.C. were supported by Grant HI17C0328 from the Korea Health Technology Research and Development Project, through the Korea Health Industry Development Institute, funded by the Ministry of Health and Welfare (2010-0018291). This work was also supported by BK21 Plus Research Fellowship from the Ministry of Education.

## Author contributions

J.C. and S.M. conceived the project. S.M. performed initial experiments and outlined the figure design. W.Y. performed the most experiments, developed computational analyses of the locomotor parameters and AIM score, bioinformatics analyses, and screened patient SNPs. H.S.R. and S.J.C. provided clinical data and method descriptions on human patient samples and SNPs. W.Y. and S.M. analyzed data. W.Y. and S.M. wrote manuscript with inputs from all authors. J.C. supervised the project.

## Competing interests

The authors declare no competing interests.
