## [Peer Review File · Communications Biology]

Reviewers' comments:

Reviewer #1 (Remarks to the Author):

In this study, the authors developed a fly model and paradigm to study the pathological mechanisms that underlie L-DOPA-induced dyskinesia (LID). LID affects 90% of Parkinson's Disease patients 9 years after long-term treatment of L-DOPA, and hence has substantial clinical impact. Since patients exhibit differential LID responses to L-DOPA, the authors sought to identify risk factors for LID pathogenesis by examining a collection of >500,00 single nucleotide polymorphisms (SNPs) from PD patients) who developed LID. To functionally screen the candidate genes, they used the genetic model *Drosophila*, whose PD models are well-established to exhibit motor symptoms rescuable by L-DOPA, and which allows for high-throughput screening.

They found that neuronal knockdown of Dop1R1 suppressed movement symptoms, while Dop1R1 overexpression alone was sufficient to trigger movement symptoms. These symptoms were completely suppressed by dADCY2 knockdown, demonstrating that dADCY2 functions downstream of Dop1R1. dADCY2 knockdown also blocked L-DOPA-induced pathological symptoms. They also identified effects of other ADCY members on both baseline and abnormal movements.

Overall, this is a well-carried out and well-controlled study. While the authors do not identify new molecular players or mechanisms, their paradigm helps to establish novel quantitative measures for studying LID in the genetic model *Drosophila*. The behavioural locomotor paradigms established here will be useful for identification and screening for other molecular players that underlie LID and other PD-related disorders.

General comments

1. References don't seem to match up, making it very difficult to properly review the paper. Please thoroughly check all of them for correct referencing, e.g.

Line 72: Should this be Reference 18 instead of 19?

Line 75: Should this be Reference 19 instead of 20?

Line 79: Should this be Reference 21 instead of 22?

Line 85: Should be Ref 23 not 24? But even 24 is not the right reference.

Also this may be the same reference mis-cited in Line 350 Ref 45

Please reference the correct primary article that shows that out of ADCY5 showed therapeutic effects on LID symptoms of PD model mice

2. Are there other *Drosophila* models of LID? Please cite them, and briefly explain the novelty of your approach compared to the other studies.

3. Please extensively proofread for spelling and grammar throughout the text, e.g. Line 344

Specific comments

1. The authors use AIM as a measure of LID-like behaviour, and the effect based on this measure is clearly very strong. However, it is unclear to me how abrupt accelerations relate to typical LID behaviours. On the other hand, speed, yawing and freezing do have some analogy to dyskinesia. Could the authors elaborate on why they choose to focus on this measure? If there isn't a strong relationship between acceleration and LID in humans, perhaps the authors can replace the term "LID-like" behaviours with "abnormal movements" instead, or try to combined these variables together into a "LID index".

2. How AIM is calculated is unclear, and since this is a key variable, please do explain it better:

Line 152: serial v_i/v_s values were logarithmized to examine whether a given $\log(v_i/v_s)$ was qualified for a threshold value (c) that had been previously determined by WT flies on control diet. The fraction of the qualified $\log(v_i/v_s)$ within the entire recording period (n) was quantified to express an AIM score using AIM scoring function $H(n)$ \diamond This is confusing, because if it is a fraction, how could AIM scores be larger than 1?

Looking at the speed plots given, I'm not clear how one can derive AIM scores as high as 8 when typically it looks like only one interval shows greatly accelerated movements. Could the authors provide one example dataset to illustrate how an AIM score is generated?

Also, since a fraction isn't used, is the recording time interval always the same length? If so, this should be clearly stated.

3. Interestingly, in their model, the abnormal movement phenotypes in some flies were reversible. Could the authors comment on how this reversibility compares to LID in humans?

4. Line 211: that Dop1R1 is necessary and sufficient for LID-like symptoms via its inhibitory role in motor control. ◊ Have there been previous studies that identified an inhibitory role in motor control for Dop1R1?

5. Were the number of UAS lines balanced between control and experiment in Figure 6? If not, this caveat should be discussed.

6. Line 295: By contrast, it had no impact either on the baseline movement or LID-like symptoms upon control and chronic L-DOPA diet (Fig. 7a-c): But, I see some difference for the AIM score caused by chronic L-DOPA. Is this not significant? Please clarify.

7. Line 302: Neuronal knockdown of dADCY3/8 markedly decreased the baseline movement ◊ Does this refer just to controls? As baseline movement for acute doesn't change, while it actually increases for chronic L-DOPA administration. Since RNAi lines may differ in efficacy, please caveat findings in this respect since RNAi efficacies were not investigated

8. Line 304: Again, I do see some suppression of AIM score in dADCY3/8 RNAi on chronic L-DOPA diet. Again, since RNAi lines may differ in efficacy, please caveat findings in this respect since RNAi efficacies were not investigated.

9. In the text, the authors alluded that ADCY2 has been already implicated in LID in mice, but the citations given do not seem correct, making it hard to follow what is previously known. The authors should comment on what this study adds to what is already known.

10. Tables S4 and S5 provide a useful summary for the rather clunky Figures 7B and 7C, and should be included as summary panels in Figure 7. Rather than arrows up and down, a heatmap could be used for better visualisation.

Minor comments

1. Line 322: "manifested in an uncontrolled fashion (involuntary)" – I believe it is difficult to determine whether a movement in the fly is voluntary, please remove

2. Line 336: "propose that the D1-like family receptor-ADCY2 signaling axis in 336 the postsynaptic dopamine neurons negatively influences fly locomotion in normal situation." By "negatively influences", do you mean slows down locomotor speed? Please rephrase.

Reviewer #2 (Remarks to the Author):

This is an interesting report by Yoon and colleagues who have established a *Drosophila* model of L-DOPA-induced dyskinesia (LID) and developed behavioral assays to track their LID-related phenotypes. Using these, the author demonstrated that dopamine 1-like receptor (Dop1R1) plays an essential role in the development of LID. They further validated a LID-associated genetic risk factor adenylyl cyclase 2 (ADCY2) that was identified by GWAS of PD patients, showing that silencing of ADCY2 expression in neurons suppresses LID symptoms in the fly induced by chronic L-DOPA administration or by Dop1R1 overexpression. The authors concluded that dysregulation in the Dop1R1-ADCY2 signaling pathway is an underlying mechanism of LID. Overall, this study has

shed important insights into the development of LID and the results obtained from their series of experiments does in large part support their conclusion. Notwithstanding the interesting report, I have some comments for the author's consideration as listed below.

1. At the lowest L-DOPA concentration (i.e. 10mM) that the authors have used, there is already a marked effect on the fly's movements. The authors have chosen this concentration to "avoid potential overdosing effects in flies". However, it is not known whether concentrations lower than 10mM (e.g. 1 or 5 mM, or lower) may be more appropriate.
2. As alterations in Dop1R1 and dADCY2 underlies LID, it begs the curiosity of what is their respective level in flies that are untreated or treated with acute or chronic dose of L-DOPA.
3. The phenotypic rescue of PINK1-B9 flies' movements by L-DOPA is as interesting as it is intriguing, because these mutant flies besides exhibiting neuronal pathology also display marked mitochondrial pathology in their muscles that promotes their movement deficits, which the authors have previously reported.
4. What is the author's interpretation on why chronic 7-day dosing of L-DOPA exacerbates AIM scores in PINK1 control flies more prominently than it did with PINK1-B9 flies? What is the implication for PD patients? Related to this, HPLC measurement of dopamine level in these flies may be helpful.
5. Notwithstanding that neuronal Dop2R knockdown had minimal impact on AIM in acute or chronic L-DOPA treated flies, it clearly restored baseline movements in these flies, but yet significantly decreased baseline movements in untreated flies. Again, the authors should explain this as some readers may be confused.

Minor

- How does the concentration of L-DOPA used in this study compared to the dose given to PD patients?
- State the concentration of DOPA in Supplementary Fig. 1 (1000mM?)
- Among the SNPs identified, how many of them leads to a change in the amino acid sequence of the translated proteins? Is there a reason why SNPs in non-coding region were excluded?
- Line 475 – "For behavioral experiments was counted as an independent measurement..." sounds incoherent.

Reviewer #3 (Remarks to the Author):

L-DOPA is considered the gold standard drug of Parkinson's disease (PD) because it is the most effective treatment of PD symptoms. However, its chronic use (>5years) is known to cause even worse motor complications called L-DOPA-induced dyskinesia (LID). Since currently there is no other promising therapeutic option for PD, LID is a serious roadblock in handling PD symptoms. The most urgent issue with relate to LID is to have better understanding on its underlying molecular and cellular mechanisms.

In this study, Yoon et al showed that LID is caused by dysregulation of D1R and adenylyl cyclase 2 (ADCY2) signaling in neurons. First, authors developed a convincing LID model of fruit fly showing abnormal involuntary movements (AIM) when high L-DOPA was chronically administered. A combined approaches of GWAS from PD patients with LID and various computational analyses were used to identify 19 candidate human genes underlying LID. Subsequently, 23 homologous fly genes were screened using their fly LID model, identifying ADCY2 as a convincing genetic link of LID.

The current study is very significant considering that LID is a serious problem in treating PD symptoms. Experiments were well designed and the scope of their approaches was very diverse. Especially I appreciated that authors presented their 'rather complicated' data in very understandable manner.

I have a few minor comments:

Minor comments:

1. It is very interesting to see 4 groups of ADCYs (Groups 1-4) on the basis of speed and AIM results with acute and chronic L-DOPA diets. Especially dADCY2 (group 2) and dADCY 9 (group 4) showed very similar AIM scores to chronic L-dopa diet with their gene knockdown. These results would be relevant to LID. It will be very helpful to discuss why results from groups 2 & 4 ADCY knockdown are similar. Authors already showed that D1R-ADCY2 signaling pathway underlies LID. Then, what is the connection of ADCY 9 to LID? Why other ADCYs (groups 1 & 3) are working differently with related to AIM scores and thus LID?
2. Authors showed another LID model with PINK1B9 fly. It is probably more interesting to study LID mechanisms which are developed from PD model. It is true that LID model here in this study is significant at any case. However, it will be interesting to add in the discussion (or relevant results section) why PINK1B9 model was not used to test D1R and ADCY2 genes with related to LID.
3. Supple Fig 10 appears to be a good working hypothesis on the basis of results of the current study. I strongly suggest to move this figure as Figure 8 into the main text, not a supplementary figure.
4. Supple Fig 9: Color coding will be very helpful to easily recognize different gene groups (human versus Drosophila, different groups of ADCYs).

My co-authors and I thank the reviewers for their constructive comments and suggestions on the manuscript. Based on their comments, the manuscript has now been fully revised as detailed below. Please see point-by-point responses **in blue**.

Responses to the concerns raised by referees:

Reviewer #1 (Remarks to the Author):

General comments

1. References don't seem to match up, making it very difficult to properly review the paper. Please thoroughly check all of them for correct referencing, e.g.

Line 72: Should this be Reference 18 instead of 19?

Line 75: Should this be Reference 19 instead of 20?

Line 79: Should this be Reference 21 instead of 22?

Line 85: Should be Ref 23 not 24? But even 24 is not the right reference.

Also this may be the same reference mis-cited in Line 350 Ref 45

Please reference the correct primary article that shows that out of ADCY5 showed therapeutic effects on LID symptoms of PD model mice

→ All references have been carefully checked and matched according to the narration of the main text.

We confirm that the primary article reporting the therapeutic effects of ADCY5 knockout on LID symptoms of PD model mice is Park *et al.*, 2014. This article is cited in the revised text, and listed in the reference accordingly (Page 5, Lines 82-83, Reference 24).

2. Are there other *Drosophila* models of LID? Please cite them, and briefly explain the novelty of your approach compared to the other studies.

→ As an additional *Drosophila* model of LID established by others, we are aware of the recent study characterizing L-DOPA-induced dyskinetic phenotypes from the *Drosophila* larvae, authored by Blosser *et al.*, 2020. In the study, the authors generated a PD model of *Drosophila* larvae expressing human mutant α -synuclein (A53T), and characterized a body bending behavior in the model as axial dyskinesia in rodent models. We have cited the study with a brief description of the new aspects of our approaches compared to that study (Pages 18-19, Lines 352-371).

There have been studies reporting L-DOPA-mediated therapeutic effects in the *Drosophila* model of PD. However, as far as we understand, our study is the first report that combines GWAS on LID-diagnosed PD patients with the *Drosophila* model of LID, and further identifies genes pathogenic for LID. The unbiased search of natural variant mutations on the patient genome allowed us to identify LID-associated genes (LAGs). Then, we crafted a computational analysis of the *Drosophila* locomotion to capture motor traits that resemble the symptoms of the

rodent model of LID including the AIMS. By examining the conserved LAGs in the fly LID paradigm, we identified ADCY2, a novel gene whose role in LID has not been previously reported. The approaches used in this study offer three main implications to the field. Our *Drosophila* model of LID provides a quick and reliable *in vivo* platform to evaluate the LID motor symptoms from various angles by using fluent genetic tools to manipulate each of the molecular and cellular components of the dopaminergic system. Our GWAS establishes a patient genomic dataset that lists natural variants potentially associated with LID, allowing us to investigate additional molecules involved in the pathogenesis of LID. Our collaborative study establishes a framework for the combined researches on LID pathogenesis between clinical data and mechanistic studies using animal models.

3. Please extensively proofread for spelling and grammar throughout the text, e.g. Line 344

→ We have corrected the grammatical errors through extensive proofreading. In particular, Line 344 has been modified (Page 19, Lines 372-374).

Specific comments

1. The authors use AIM as a measure of LID-like behaviour, and the effect based on this measure is clearly very strong. However, it is unclear to me how abrupt accelerations relate to typical LID behaviours. On the other hand, speed, yawing and freezing do have some analogy to dyskinesia. Could the authors elaborate on why they choose to focus on this measure? If there isn't a strong relationship between acceleration and LID in humans, perhaps the authors can replace the term "LID-like" behaviours with "abnormal movements" instead, or try to combine these variables together into a "LID index".

→ We thank the reviewer for the suggestion. One of the hallmark symptoms of LID in humans is the occurrence of a spectrum of hyperkinetic movements, including chorea, dystonia and myoclonus, which often appear abruptly in an uncontrolled fashion from a stationary posture (Fahn, 2000). In a rodent model of LID, these hyperkinetic movements are collectively translated into the abnormal involuntary movement (AIM) as a well-established LID parameter (Keber *et al.*, 2015). We focused on those abrupt accelerations of fly movements that are irregular and deviated from the regular movement at the near time point, as the *Drosophila* version of AIM. Systematical analysis of the L-DOPA-induced AIM of the fly revealed that they are highly correlated with the dose and duration of L-DOPA treatment. Moreover, the L-DOPA-induced fly AIM is altered by manipulation of the components in the dopaminergic system, indicating that the fly AIM is associated with the dopaminergic signaling pathway underlying LID pathogenesis. Therefore, we infer that the abrupt acceleration of the fly is analogous to the hyperkinetic movements in the rodent models and humans. To elaborate, we have modified our text about the hyperkinetic movement (Page 8, Lines 138-145, Reference 6-8, 34).

2. How AIM is calculated is unclear, and since this is a key variable, please do explain it better: Line 152: serial vi/vs values were logarithmized to examine whether a given $\log(v_i/v_s)$ was qualified for a threshold value (c) that had been previously determined by WT flies on control

diet. The fraction of the qualified $\log(v_i/v_s)$ within the entire recording period (n) was quantified to express an AIM score using AIM scoring function($H(n)$) \diamond This is confusing, because if it is a fraction, how could AIM scores be larger than 1?

→ We apologize for the confusion. To clarify, the AIM score (H) is the sum of 0 or 1 given to each frame (9,018 frames total/ 5 min) during the entire recording based on whether the value of $\log(v_i/v_s)$ at a frame qualifies the threshold value (c). If the value of $\log(v_i/v_s)$ at a frame is larger than c , 1 is given. If the value of $\log(v_i/v_s)$ is smaller than c , 0 is given. For example, if a fly showed the value of $\log(v_i/v_s)$ larger than c at seven different frames, then the AIM score of the fly is 7. For better visualization of data, the AIM score of control flies is normalized to be 0, where the threshold value (c) is 0.4. To help understand this better, we modified the description in the text and the Method section (Page 9, Lines 155-161, and Page 23, Lines, 440-447). Additionally, we have provided a new Supplementary Data 1 to show raw data used to make the AIM graphs in Fig. 2b-d.

Looking at the speed plots given, I'm not clear how one can derive AIM scores as high as 8 when typically it looks like only one interval shows greatly accelerated movements. Could the authors provide one example dataset to illustrate how an AIM score is generated?

→ Please see our response above for the AIM scoring logic. The AIM score (H) is the sum of 0 or 1 given to each frame (9,018 frames total/ 5 min) during the entire recording based on whether the value of $\log(v_i/v_s)$ at a frame qualifies the threshold value (c). As requested by the reviewer, we provide real data points in the new Supplementary Data 1.

Also, since a fraction isn't used, is the recording time interval always the same length? If so, this should be clearly stated.

→ Please see our response above. We obtained the AIM score from 5-minute video with 9,018 frames. To clarify, we have modified the description about the AIM scoring logic in the description of the text and the Method section (Page 8-9, Lines 155-161, and Page 23, Lines, 440-447).

3. Interestingly, in their model, the abnormal movement phenotypes in some flies were reversible. Could the authors comment on how this reversibility compares to LID in humans?

→ We thank the reviewer for the interesting question. To clarify, we believe that the reversible phenotype pointed out by the reviewer is meant for the restored AIM score from a portion of *PINK^{B9}* flies that showed the mild AIM score upon the prior L-DOPA diet. We want to ensure that the AIM score is too low to be justified for a statistical significance. The effect of L-DOPA early in the course of the treatment is usually beneficial and long lasting with less likelihood of the dyskinesia occurrence, and the dyskinetic symptoms may be mild in the early phase of LID (Fox and Lang, 2008; Pandey and Srivanitchapoom, 2017). Given the acute L-DOPA treatment largely beneficial to the PD symptoms (speed and trajectory) of the mutant fly, reversibility here may represent the therapeutic effect during the early treatment, and the mild AIM score from the

L-DOPA treatment may indicate very early phase of LID progression. It is generally appreciated that once LID develops it is difficult to reverse. Low dosing of L-DOPA can be an option, while the dyskinetic symptoms may still be persistent. Pulsatile stimulation of the receptors in the dopaminergic pathway is thought to underlie the abrupt appearance of dyskinetic symptoms (Calabresi *et al.*, 2010). This offers a clinical opportunity that L-DOPA dosing complemented with pharmacological intervention of the receptors underlying the pulsatility would be a way to reverse the symptoms. Interestingly, a recent study reported that viral delivery-guided inhibition of CaV1.3 channels in the striatum reversed the dyskinetic symptoms of animal models with severe LID. Importantly, the motor benefit from low dosing of L-DOPA was still preserved with the inhibition of the CaV1.3 channels, suggesting that this ion channel may underlie the reversibility of LID pathogenesis, independently of the dyskinesia-inducing mechanism (Steece-Collier *et al.*, 2019).

4. Line 211: that Dop1R1 is necessary and sufficient for LID-like symptoms via its inhibitory role in motor control. Have there been previous studies that identified an inhibitory role in motor control for Dop1R1?

→ Consistent with our observation, multiple literatures previously reported that disruption of Dop1R1 expression increases motor activities (Kim *et al.*, 2016; Lebestky *et al.*, 2009; Silva *et al.*, 2020). By contrast, it remained unclear whether Dop1R1 expression by itself would decrease locomotion. Our study is newly reporting that neuronal overexpression of Dop1R1 decreases locomotion. Together with Dop1R1 knockdown/ knockout data, our observations suggest that Dop1R1 expression is inversely correlated with locomotion.

5. Were the number of UAS lines balanced between control and experiment in Figure 6? If not, this caveat should be discussed.

→ We now discuss this point in “Fly stocks and husbandry” section of the Method (Page 22, Lines 428-429).

6. Line 295: By contrast, it had no impact either on the baseline movement or LID-like symptoms upon control and chronic L-DOPA diet (Fig. 7a-c): But, I see some difference for the AIM score caused by chronic L-DOPA. Is this not significant? Please clarify.

→ We apologize for the confusion. As pointed out by the reviewer, we confirm that the flies with RNAi knockdown of ADCY1 and ADCY5 showed mildly decreased AIM scores compared to WT flies. Thus, we have modified the sentence accordingly (Page 15, Lines 304-306).

7. Line 302: Neuronal knockdown of dADCY3/8 markedly decreased the baseline movement. Does this refer just to controls? As baseline movement for acute doesn't change, while it actually increases for chronic L-DOPA administration. Since RNAi lines may differ in efficacy, please caveat findings in this respect since RNAi efficacies were not investigated

→ We want to emphasize that the ADCY RNAi lines used in our study have already been validated by various sources of data including the cited articles and public *Drosophila* RNAi database (Duvall and Taghert, 2012; 2013; Onur *et al.*, 2021; Perkins *et al.*, 2015; https://www.flyrnai.org/cgi-bin/RSVP_search.pl). Therefore, we addressed this point by modifying the text with referenced (Pages 15, Lines 302-303, Reference 43-46).

8. Line 304: Again, I do see some suppression of AIM score in dADCY3/8 RNAi on chronic L-DOPA diet. Again, since RNAi lines may differ in efficacy, please caveat findings in this respect since RNAi efficacies were not investigated.

→ Please see our response above.

9. In the text, the authors alluded that ADCY2 has been already implicated in LID in mice, but the citations given do not seem correct, making it hard to follow what is previously known. The authors should comment on what this study adds to what is already known.

→ This comment has stemmed from the error in the ADCY5-related references. To clarify, ADCY5, but not ADCY2, has been implicated in the mouse model of LID. We have discussions about the ADCY5 studies with them referenced correctly in the revised manuscript (Page 5, Lines 82-83, and Page 19, Lines 372-378, Reference 24). In our study, we report that the fly ADCY2 functions downstream of Dop1R1 (a homolog of D1 receptor) in *Drosophila* model of LID. As far as we know, this is the first study reporting the role of ADCY2 on LID symptoms and showing that ADCY2 functions downstream of D1-like receptor to mediate LID pathogenesis.

10. Tables S4 and S5 provide a useful summary for the rather clunky Figures 7B and 7C, and should be included as summary panels in Figure 7. Rather than arrows up and down, a heatmap could be used for better visualization.

→ We agree that the bar graphs are not satisfactory to efficiently convey our conclusions. We have moved the summary panels to Figure 7 for better visualization of data and a new panel with heatmaps has been added in Figure 7 (Fig. 7e and f).

Minor comments

1. Line 322: “manifested in an uncontrolled fashion (involuntary)” – I believe it is difficult to determine whether a movement in the fly is voluntary, please remove

→ We removed the sentence.

2. Line 336: “propose that the D1-like family receptor-ADCY2 signaling axis in 336 the postsynaptic dopamine neurons negatively influence fly locomotion in normal situation.” By “negatively influences”, do you mean slows down locomotor speed? Please rephrase.

→ As suggested, “negatively influences” has been changed to “slows down locomotor speed” (Page 17, Line 344).

Reviewer #2 (Remarks to the Author):

1. At the lowest L-DOPA concentration (i.e. 10 mM) that the authors have used, there is already a marked effect on the fly's movements. The authors have chosen this concentration to "avoid potential overdosing effects in flies". However, it is not known whether concentrations lower than 10 mM (e.g. 1 or 5 mM, or lower) may be more appropriate.

→ We thank the reviewer for raising the important point. Previous studies have used various concentrations of L-DOPA ranging from 0.1 mM to 50.71 mM for their experimental purposes. For example, Lanno *et al.* used 50.71 mM for their analysis on L-DOPA-induced genomic alterations in flies, Niens *et al.* used 5.07 mM in their studies on the effect of dopaminergic signaling on serotonergic innervation in the brain, and we have previously used 1 mM and observed about 70% rescue effect on motor abilities of the PD model flies with *parkin* knockout. More recently, Blosser *et al.* used 5 and 10 mM of L-DOPA to show that these "higher" concentrations induce a fly larval version of LID symptoms that were not efficiently observed from 0.1 mM of L-DOPA. Similarly, in our setup, 10 mM allowed us to examine reliable LID parameters in *Drosophila*.

2. As alterations in Dop1R1 and dADCY2 underlies LID, it begs the curiosity of what is their respective level in flies that are untreated or treated with acute or chronic dose of L-DOPA.

→ This is very exciting question that we have been also interested. In unpublished experiments, we sought to monitor expression pattern, expression level and activity of the dopamine receptor, ADCY2 and their downstream products such as cyclic AMP (cAMP) by means of tagged proteins, antibodies and cAMP indicators. Then, we sought to compare alterations of these molecules in control flies and the manipulated fly in which the dopaminergic circuit artificially activated by neuronal stimulation or L-DOPA treatment. These experiments are ongoing with a hope that we find the physiological link between L-DOPA treatment and the pathogenic molecules in the Dop1R1-ADCY2 axis. We have been pursuing this idea further to mature the work for a publication. However, we believe that this would be beyond the scope of our current study.

3. The phenotypic rescue of PINK1-B9 flies' movements by L-DOPA is as interesting as it is intriguing, because these mutant flies besides exhibiting neuronal pathology also display marked mitochondrial pathology in their muscles that promotes their movement deficits, which the authors have previously reported.

→ As commented by the reviewer, *PINK1^{B9}* mutant flies possess not only the neuropathology, but also mitochondrial deficits in the muscle, remaining an open question – in which tissue L-DOPA affects to rescue. There are two possible scenarios in which the rescue effect is produced by 1) L-DOPA-mediated improvement of mitochondrial quality in the muscle, or 2) L-DOPA-mediated alleviation of the dopaminergic impairment in the nervous system of the PD model flies.

In our previous study, we have shown that L-DOPA diet rescues the motor defects of the *parkin* PD model flies through the subset-specific improvement of the neuronal deficit in the brain dopaminergic neurons (Cha *et al.*, 2005). Since Parkin is the downstream effector of PINK1, it is highly likely that the locomotor rescue in *PINK1^{B9}* flies was mediated through the improvement of dopaminergic deficits in the brain. Supporting the view, ADCY2 knockdown in neurons blocked neuronal Dop1R1-mediated LID symptoms. Together, these data suggest that the phenotypic rescue in *PINK1^{B9}* flies is mediated through the neuronal pathway.

4. What is the author's interpretation on why chronic 7-day dosing of L-DOPA exacerbates AIM scores in PINK1 control flies more prominently than it did with PINK1-B9 flies? What is the implication for PD patients? Related to this, HPLC measurement of dopamine level in these flies may be helpful.

→ It is an interesting observation that chronic dosing of L-DOPA exacerbated the AIM score in the control than *PINK1^{B9}* flies. The parsimonious interpretation of this observation is that excessive dopamine supplies accompanying overstimulation of the dopamine receptors and downstream signaling components could have contributed to the exacerbated phenotypes in the control flies. The control flies do not have the demand for exogenous dopamine supply, while *PINK1^{B9}* flies are in demand of the exogenous dopamine due to the genetically impaired dopaminergic supply. We can imagine that chronic dosing of L-DOPA in the control flies creates extra supply of dopamine that causes exaggerated stimulation of the postsynaptic dopamine signaling resulting in multiple pathological changes in downstream neurons and molecules. This idea supports the clinical observation that the use of higher dose of L-DOPA tend to advance LID symptoms. Measurement of dopamine levels in these flies would be informative in understanding the pathophysiological mechanism. More informative would be to measure local changes of dopamine levels at the site of its action (neurons) or dopamine receptor activity in these control flies in comparison to *PINK1^{B9}* flies. If we can customize dosing of L-DOPA based on information from these physiological measurements, we would be able to reduce the risk of LID pathogenesis by precisely controlling the amount, frequency and timing of L-DOPA treatment.

5. Notwithstanding that neuronal Dop2R knockdown had minimal impact on AIM in acute or chronic L-DOPA treated flies, it clearly restored baseline movements in these flies, but yet significantly decreased baseline movements in untreated flies. Again, the authors should explain this as some readers may be confused.

→ We agree with the reviewer's point. Thus, we have modified the text to include these observations (Page 11, Lines 212-216).

Minor

- How does the concentration of L-DOPA used in this study compared to the dose given to PD patients?

→ According to a previous literature, PD patients dosed with L-DOPA ranging from 27.9 mM - 99.9 mM showed 5 ~ 104% improvements in their locomotion symptoms (Johnels *et al.*, 1993). In another study, HPLC measurements from “good responder” PD patients who took 375 – 750 mg/day showed 0.3 nM - 11.4 nM of plasma L-DOPA after 90 minutes (Dutton *et al.*, 1993). In our experiments, we housed 10 flies with 2 mg of L-DOPA (0.2 mg/fly and 1 mM/fly) either for a day or 7 days as the acute or chronic treatment. Apparently, direct comparison of the L-DOPA dose between human and the fly is limited, because humans have completely different L-DOPA pharmacokinetics in the body compared to the fly for the major difference in absorption, dopamine converting enzymes, circulation, etc (Contin and Martinelli, 2010). Nevertheless, our L-DOPA dosing appears generally below the concentration used in PD patients.

- State the concentration of DOPA in Supplementary Fig. 1 (1000 mM?)

→ Except for Fig 1, all concentrations used were 10 mM, unless otherwise indicated. We have addressed this comment in Supplementary Fig. 1.

- Among the SNPs identified, how many of them leads to a change in the amino acid sequence of the translated proteins? Is there a reason why SNPs in non-coding region were excluded?

→ In our data analysis, we included not only the SNPs on protein coding regions, but also the ones on non-coding regions. From 210 SNPs with $p < 10^{-6}$ and their loci confirmed on the open reading frame, we found that 80.48% (169) of the SNPs is distributed on the introns, including one splice site mutation. From 19.52% (41) of the SNPs on the exons, 17.14% (36) causes nonsynonymous substitutions that result in missense mutations, and 2.38% (5) produces synonymous substitution.

- Line 475 – “For behavioral experiments was counted as an independent measurement...” sounds incoherent.

→ We have modified the sentence. (Page 27, Lines 524-525).

Reviewer #3 (Remarks to the Author):

Minor comments:

1. It is very interesting to see 4 groups of ADCYs (Groups 1-4) on the basis of speed and AIM results with acute and chronic L-DOPA diets. Especially dADCY2 (group 2) and dADCY 9 (group 4) showed very similar AIM scores to chronic L-dopa diet with their gene knockdown. These results would be relevant to LID. It will be very helpful to discuss why results from groups 2 & 4 ADCY knockdown are similar. Authors already showed that D1R-ADCY2 signaling pathway underlies LID. Then, what is the connection of ADCY 9 to LID? Why other ADCYs (groups 1 & 3) are working differently with related to AIM scores and thus LID?

→ We thank the reviewer for bringing the outstanding question. We have updated the discussion about ADCYs with previous studies referenced (Pages 19-20, Lines 386-403). Depending on which upstream and downstream signaling components are coupled with ADCYs, how strongly the signaling cascade persists, and which cells they are expressed in, there seem to be temporal and spatial dimensions in which these ADCYs interplay in mediating LID pathogenesis. Without knowing *in vivo* characteristics of ADCYs associated with L-DOPA treatment in respect to their interacting partners, expression patterns, expression levels and activity, this discussion would be limited. ADCY5 is the best studied ADCY member for its role in a variety of physiologies including cardiac homeostasis, longevity and LID. In *Drosophila*, there is a report that examined the role of ADCYs in circadian behaviors, showing that AC3 (ADCY3) is involved in the anticipatory behavior in circadian rhythm by functioning in a subset of clock cells in the brain (Duvall and Taghert, 2012). These studies suggest that there may be temporal and spatial specificity of ADCYs in different physiologies and behaviors. Using a genetically tractable model organism like *Drosophila* which permits cell-specific manipulations and recordings of ADCYs, it would be possible to further address the temporal and spatial expression pattern of ADCYs in LID-associated tissues and their functions in the L-DOPA-induced motor response.

The major phenotypical difference between ADCY group 2 (ADCY2) and 4 (ADCY9) is that knockdown of ADCY2, but not ADCY9, causes increase of the baseline movement in the absence of L-DOPA, completely phenocopying knockdown/ knockout of Dop1R1. Together, we propose that ADCY2 and ADCY9 may share a common cellular and molecular pathway in the pathogenesis of LID symptoms, while the role of ADCY9 in the baseline movement could be modulatory (dispensable) and independent of Dop1R1. We discuss these ideas in the result section (Page 16, Lines 317-322).

2. Authors showed another LID model with *PINK1^{B9}* fly. It is probably more interesting to study LID mechanisms which are developed from PD model. It is true that LID model here in this study is significant at any case. However, it will be interesting to add in the discussion (or relevant results section) why *PINK1^{B9}* model was not used to test D1R and ADCY2 genes with related to LID.

→ We completely agree with the comment about using *PINK1^{B9}* mutant flies. However, unfortunately, there is a major technical limitation for us to use *PINK1^{B9}* flies. These flies are

short lived and possess deficits in reproduction: homozygous male flies are sterile, and female flies are lethal (Park *et al.*, 2006). Given the number of transgenic elements to insert into the *PINK1^{B9}* background for the experiments, we were unable to perform D1R and ADCY2 manipulations in the PD model flies using currently available techniques. We have added discussions about this technical challenge (Page 11, Lines 212-216).

3. Supple Fig 10 appears to be a good working hypothesis on the basis of results of the current study. I strongly suggest to move this figure as Figure 8 into the main text, not a supplementary figure.

→ We thank the reviewer for the suggestion. Supplementary Fig. 10 has been moved to Fig. 8 as suggested.

4. Supple Fig 9: Color coding will be very helpful to easily recognize different gene groups (human versus *Drosophila*, different groups of ADCYs).

→ We have color-coded the gene groups for better visualization as suggested.

References

- Blosser, J.A., Podolsky, E., and Lee, D. (2020). L-DOPA-Induced Dyskinesia in a Genetic *Drosophila* Model of Parkinson's Disease. *Exp Neurobiol* 29, 273-284. 10.5607/en20028.
- Calabresi, P., Di Filippo, M., Ghiglieri, V., Tambasco, N., and Picconi, B. (2010). Levodopa-induced dyskinesias in patients with Parkinson's disease: filling the bench-to-bedside gap. *Lancet Neurol* 9, 1106-1117. 10.1016/S1474-4422(10)70218-0.
- Cha, G.H., Kim, S., Park, J., Lee, E., Kim, M., Lee, S.B., Kim, J.M., Chung, J., and Cho, K.S. (2005). Parkin negatively regulates JNK pathway in the dopaminergic neurons of *Drosophila*. *Proc Natl Acad Sci U S A* 102, 10345-10350. 10.1073/pnas.0500346102.
- Contin, M., and Martinelli, P. (2010). Pharmacokinetics of levodopa. *J Neurol* 257, S253-261. 10.1007/s00415-010-5728-8.
- Dutton, J., Copeland, L.G., Playfer, J.R., and Roberts, N.B. (1993). Measuring L-dopa in plasma and urine to monitor therapy of elderly patients with Parkinson disease treated with L-dopa and a dopa decarboxylase inhibitor. *Clin Chem* 39, 629-634.
- Duvall, L.B., and Taghert, P.H. (2012). The circadian neuropeptide PDF signals preferentially through a specific adenylate cyclase isoform AC3 in M pacemakers of *Drosophila*. *PLoS Biol* 10, e1001337. 10.1371/journal.pbio.1001337.
- Duvall, L.B., and Taghert, P.H. (2013). E and M circadian pacemaker neurons use different PDF receptor signalosome components in *drosophila*. *J Biol Rhythms* 28, 239-248. 10.1177/0748730413497179.
- Fahn, S. (2000). The spectrum of levodopa-induced dyskinesias. *Ann Neurol* 47, S2-9; discussion S9-11.
- Fox, S.H., and Lang, A.E. (2008). Levodopa-related motor complications--phenomenology. *Mov Disord* 23 Suppl 3, S509-514. 10.1002/mds.22021.
- Johnels, B., Ingvarsson, P.E., Holmberg, B., Matousek, M., and Steg, G. (1993). Single-dose L-dopa response in early Parkinson's disease: measurements with optoelectronic recording technique. *Mov Disord* 8, 56-62. 10.1002/mds.870080111.
- Keber, U., Kliez, M., Carlsson, T., Oertel, W.H., Weihe, E., Schafer, M.K., Hoglinger, G.U., and Depboylu, C. (2015). Striatal tyrosine hydroxylase-positive neurons are associated with L-DOPA-induced dyskinesia in hemiparkinsonian mice. *Neuroscience* 298, 302-317. 10.1016/j.neuroscience.2015.04.021.
- Kim, S., Tellez, K., Buchan, G., and Lebestky, T. (2016). Fly Stampede 2.0: A Next Generation Optomotor Assay for Walking Behavior in *Drosophila Melanogaster*. *Front Mol Neurosci* 9, 148. 10.3389/fnmol.2016.00148.

- Lanno, S.M., Lam, I., Drum, Z., Linde, S.C., Gregory, S.M., Shimshak, S.J., Becker, M.V., Brew, K.E., Budhiraja, A., Carter, E.A., *et al.* (2019). Genomics Analysis of L-DOPA Exposure in *Drosophila sechellia*. *G3 (Bethesda)* 9, 3973-3980. 10.1534/g3.119.400552.
- Lebestky, T., Chang, J.S., Dankert, H., Zelnik, L., Kim, Y.C., Han, K.A., Wolf, F.W., Perona, P., and Anderson, D.J. (2009). Two different forms of arousal in *Drosophila* are oppositely regulated by the dopamine D1 receptor ortholog DopR via distinct neural circuits. *Neuron* 64, 522-536. 10.1016/j.neuron.2009.09.031.
- Niens, J., Reh, F., Coban, B., Cichewicz, K., Eckardt, J., Liu, Y.T., Hirsh, J., and Riemensperger, T.D. (2017). Dopamine Modulates Serotonin Innervation in the *Drosophila* Brain. *Front Syst Neurosci* 11, 76. 10.3389/fnsys.2017.00076.
- Onur, T.S., Laitman, A., Zhao, H., Keyho, R., Kim, H., Wang, J., Mair, M., Wang, H., Li, L., Perez, A., *et al.* (2021). Downregulation of glial genes involved in synaptic function mitigates Huntington's disease pathogenesis. *Elife* 10. 10.7554/eLife.64564.
- Pandey, S., and Srivanitchapoom, P. (2017). Levodopa-induced Dyskinesia: Clinical Features, Pathophysiology, and Medical Management. *Ann Indian Acad Neurol* 20, 190-198. 10.4103/aian.AIAN_239_17.
- Park, H.Y., Kang, Y.M., Kang, Y., Park, T.S., Ryu, Y.K., Hwang, J.H., Kim, Y.H., Chung, B.H., Nam, K.H., Kim, M.R., *et al.* (2014). Inhibition of adenylyl cyclase type 5 prevents L-DOPA-induced dyskinesia in an animal model of Parkinson's disease. *J Neurosci* 34, 11744-11753. 10.1523/JNEUROSCI.0864-14.2014.
- Park, J., Lee, S.B., Lee, S., Kim, Y., Song, S., Kim, S., Bae, E., Kim, J., Shong, M., Kim, J.M., and Chung, J. (2006). Mitochondrial dysfunction in *Drosophila* PINK1 mutants is complemented by parkin. *Nature* 441, 1157-1161. 10.1038/nature04788.
- Perkins, L.A., Holderbaum, L., Tao, R., Hu, Y., Sopko, R., McCall, K., Yang-Zhou, D., Flockhart, I., Binari, R., Shim, H.S., *et al.* (2015). The Transgenic RNAi Project at Harvard Medical School: Resources and Validation. *Genetics* 201, 843-852. 10.1534/genetics.115.180208.
- Silva, B., Hidalgo, S., and Campusano, J.M. (2020). Dop1R1, a type 1 dopaminergic receptor expressed in Mushroom Bodies, modulates *Drosophila* larval locomotion. *PLoS One* 15, e0229671. 10.1371/journal.pone.0229671.
- Steece-Collier, K., Stancati, J.A., Collier, N.J., Sandoval, I.M., Mercado, N.M., Sortwell, C.E., Collier, T.J., and Manfredsson, F.P. (2019). Genetic silencing of striatal CaV1.3 prevents and ameliorates levodopa dyskinesia. *Mov Disord* 34, 697-707. 10.1002/mds.27695.

https://www.flyrnai.org/cgi-bin/RSVP_search.pl

REVIEWERS' COMMENTS:

Reviewer #2 (Remarks to the Author):

The revised manuscript is a significantly improved version and I am generally satisfied with the response provided by the authors to my comments.

Reviewer #3 (Remarks to the Author):

I thoroughly read the revised MS and their responses to reviewers' critiques. Authors have adequately addressed all reviewers' concerns. I do not have any additional comments on the revised version.